# LMask: Learn to Solve Constrained Routing Problems with Lazy Masking

**Tianyou Li**[1*]   **Haijun Zou**[2*]   **Jiayuan Wu**[3]   **Zaiwen Wen**[1†]

[1]Beijing International Center for Mathematical Research, Peking University, China
[2]Academy of Mathematics and Systems Science, Chinese Academy of Sciences, China
[3]Department of Statistics and Data Science, University of Pennsylvania, United States

## Abstract

Routing problems are canonical combinatorial optimization tasks with wide-ranging applications in logistics, transportation, and supply chain management. However, solving these problems becomes significantly more challenging when complex constraints are involved. In this paper, we propose LMask, a novel learning framework that utilizes dynamic masking to generate high-quality feasible solutions for constrained routing problems. LMask introduces the LazyMask decoding method, which lazily refines feasibility masks with the backtracking mechanism. In addition, it employs the refinement intensity embedding to encode the search trace into the model, mitigating representation ambiguities induced by backtracking. To further reduce sampling cost, LMask sets a backtracking budget during decoding, while constraint violations are penalized in the loss function during training to counteract infeasibility caused by this budget. We provide theoretical guarantees for the validity and probabilistic optimality of our approach. Extensive experiments on the traveling salesman problem with time windows (TSPTW) and TSP with draft limits (TSPDL) demonstrate that LMask achieves state-of-the-art feasibility rates and solution quality, outperforming existing neural methods.

## 1 Introduction

Routing problems form a fundamental class of combinatorial optimization (CO) problems, encompassing the traveling salesman problem (TSP), vehicle routing problem (VRP), and their numerous variants. These problems frequently arise in practical domains such as logistics (Konstantakopoulos et al., 2022), transportation (Díaz-Parra et al., 2014), and supply chain management Duan et al. (2020), where the objective is to determine optimal routes while satisfying a variety of constraints, such as vehicle capacities, draft limits, time windows, or precedence requirements. The presence of these constraints significantly increases the problem complexity, as they must all be considered when optimizing the distance, time, or transport cost. Integer linear programming (ILP) provides a rigorous framework for modeling routing problems while incorporating constraints explicitly (Dantzig et al., 1954; Dantzig & Ramser, 1959). It guarantees optimal solutions under feasible conditions; however, as the number and complexity of constraints grow, the computational cost of solving ILP models increases exponentially. This makes ILP computationally intractable especially for large-scale instances with complex constraints (Cook, 2015).

To address the computational challenges posed by complex constraints, heuristic methods are commonly adopted. These methods efficiently generate high-quality solutions using approximations and relaxation strategies to handle constraints. Prominent heuristic approaches, such as the Lin-Kernighan-Helsgaun (LKH) heuristic (Helsgaun, 2017), fast iterated local optimization (Accorsi & Vigo, 2021), and hybrid genetic search (Vidal, 2022; Wouda et al., 2024), have demonstrated effectiveness across various constrained routing problem variants. Nonetheless, applying these heuristics often requires significant attention to constraint modeling, algorithm customization, and parameter tuning. This

---

*Equal contribution, alphabetical order
†Corresponding author (email: wenzw@pku.edu.cn)

highlights the necessity for expert knowledge to adapt methods for specific problem settings, as well as the complexity of constraint handling in CO problems.

In recent years, machine learning for CO approaches have been developed to tackle the difficulty in solving CO problems. Neural constructive solvers are learning-based methods designed to construct a complete route by adding a node to the current partial route sequentially. Early works introduce the pointer network to generate the optimal solution to TSP (Vinyals et al., 2015; Bello et al., 2017) and VRP (Nazari et al., 2018) in an auto-regressive way. The attention-based model (AM) (Kool et al., 2018) is a fundamental work in this line of research, which adopts a transformer-based model architecture. Building on AM, many subsequent innovations, including dynamic embedding (Peng et al., 2020; Luo et al., 2023), symmetry utilization (Kwon et al., 2020; Kim et al., 2022) and posterior search (Hottung et al., 2022; Choo et al., 2022), have been proposed to facilitate the performance. Their decoders, utilizing attention mechanisms, generate the solution incrementally by selecting nodes in an auto-regressive manner. Masking techniques are adopted to exclude nodes that do not satisfy the constraints. Additionally, there are several works studying foundation models for VRPs, such as MVMoE (Zhou et al., 2024), Routefinder (Berto et al., 2024) and CaDA (Li et al., 2024).

Most neural methods for routing problems handle constraints by employing the masking mechanism, which excludes actions directly leading to infeasible solutions during construction (Kool et al., 2018; Huang et al., 2025; Kwon et al., 2020; Joshi et al., 2019; Luo et al., 2023). This sequential construction is valid due to the property of tail recursion: after applying a series of construction steps, the remaining tail subproblem becomes a smaller instance of the original CO problem, as discussed in (Drakulic et al., 2023). The construction process implicitly assumes that the tail subproblem is always feasible. This property is present in ordinary routing problems, allowing feasible solutions to be generated node by node. However, for route problems with complex constraints, feasibility issues pose a major challenge during the sequential generation process. To tackle the feasibility difficulty, recent neural approaches have developed diverse strategies. Kool et al. (2022) propose DPDP, which combines learned neural heuristics with dynamic programming algorithms to handle hard constraints. Ma et al. (2023) present the neural k-opt solver for TSP and CVRP, which learns the search process with feasibility-related features and guided infeasible region exploration scheme. Chen et al. (2024) develop a multi-step look-ahead method tailored for TSPTW, incorporating problem-specific features and a large supervised learning dataset. Bi et al. (2024) propose a proactive infeasibility prevention (PIP) framework based on preventative infeasibility (PI) masking, learnable decoders, and adaptive strategies to advance neural methods.

In the previous learning methods, the one-pass forward sequence construction framework (Kool et al., 2018; Kwon et al., 2020; Luo et al., 2023; Bi et al., 2024) limits the model's ability to handle complex constraints. This auto-regressive approach builds a solution step-by-step in an irreversible manner, where each action is based solely on the partial solution generated so far. Although lookahead strategies attempt to mitigate the inflexibility by exploring the following feasible actions, they are computationally expensive with no guarantee of finding a feasible solution. Furthermore, to enforce these constraints during generation, such methods heavily rely on a masking mechanism that lacks a systematic mathematical explanation. This motivates us to develop a more effective constructive framework with a distinct decoding method, enabling learning over more general constraints in routing problems. Therefore, we develop a novel learning framework, called **LMask**, to handle the feasibility issue in solving routing problems with complex constraints. It explains how machine learning can solve NP-hard problems in an end-to-end manner, offering a theoretically guaranteed approach to generating feasible solutions. The name **LMask** on one hand represents the "LazyMask" decoding algorithm, and on the other hand signifies using the "mask" refinement mechanism to "learn" the routing problems with complex constraints. The overall LMask framework is illustrated in Figure 1 at the end of Section 3. Our main contributions are summarized as follows.

1) **Mechanism innovation.** We propose the LazyMask decoding algorithm, which lazily updates feasibility masks through the backtracking mechanism, enabling efficient generation of feasible solutions with theoretical guarantees. To mitigate representation ambiguities, the refinement intensity embedding is further employed to integrate information about the search trace into our model.

2) **Theoretical guarantee.** We present a systematic explanation of the masking mechanism's role in routing problems, which fundamentally guides the design of the LMask framework. Based on our mathematical derivation, rigorous theoretical guarantees demonstrate the validity of the LazyMask algorithm and the effectivity of LMask's probabilistic model with entropy regularization.

3) **Experimental outperformance.** Comprehensive experiments on TSPTW and TSPDL demonstrate that LMask achieves significantly higher solution feasibility and smaller objective gaps than other neural constructive methods with comparable runtime, showing state-of-the-art performance. Notably, a nearly 0.0% infeasibility rate is observed for LMask on synthetic TSPTW datasets, highlighting its effectiveness in handling complex constraints.

## 2 PRELIMINARIES FOR ROUTING PROBLEMS

### 2.1 A UNIFIED FORMULATION

In the context of end-to-end learning, we typically do not use mixed-integer programming for modeling routing problems due to its high computational complexity. Instead, we consider a unified formulation of routing problems. Let $V := \{0, 1, \ldots, n\}$ denote the set of nodes and $\Pi := V^T$ represent the sequence space containing all possible routes of length $T$. The length $T$ is always $n + 1$ when a routing problem requires each node to be visited exactly once. A wide range of routing problems can be expressed using the following formulation:

$$\min_{\pi \in \Pi} \quad f(\pi; \mathcal{P}), \quad \text{s.t.} \quad c(\pi; \mathcal{P}) \leq 0, \ d(\pi; \mathcal{P}) = 0, \tag{1}$$

where $\mathcal{P}$ represents the problem instance, $c(\pi; \mathcal{P})$ and $d(\pi; \mathcal{P})$ represent the hard constraints imposed on the route $\pi$. $d(\pi; \mathcal{P}) = 0$ can be the visit constraints that each node is exactly visited once. $c(\pi; \mathcal{P}) \leq 0$ can represent time window constraints, draft limit constraints, etc. More details of the formulation are shown in Appendix A.

### 2.2 A DISTRIBUTION APPROXIMATION VIEW

In this section, we provide a novel view from the perspective of distribution approximation to solve problem (1). Let $\Pi^*$ be the optimal solution set and $f^*(\mathcal{P})$ be the optimal objective value for a given instance $\mathcal{P}$. The search for optimal solutions can be framed as identifying the target distribution:

$$q^*(\pi; \mathcal{P}) := \frac{1}{|\Pi^*|} \mathbb{1}_{\Pi^*}(\pi) = \begin{cases} \frac{1}{|\Pi^*|}, & \pi \in \Pi^*, \\ 0, & \pi \notin \Pi^*. \end{cases}$$

Since the optimal solutions cannot be determined in advance, the target distribution is computationally inaccessible. However, it can be approximated by a family of constrained Gibbs distributions:

$$q_\lambda(\pi; \mathcal{P}) := \frac{1}{Z_\lambda} \exp\left(-\frac{f(\pi; \mathcal{P}) - f^*(\mathcal{P})}{\lambda}\right) \mathbb{1}_C(\pi),$$

where $C := \{\pi \in \Pi : c(\pi; \mathcal{P}) \leq 0, d(\pi; \mathcal{P}) = 0\}$ is the feasible set of problem (1) and $Z_\lambda := \sum_{\pi \in C} \exp\left(-(f(\pi; \mathcal{P}) - f^*(\mathcal{P}))/\lambda\right)$. It is also known as an energy-based model and has a profound impact in deep learning (LeCun et al., 2006; Song & Kingma, 2021). It can be readily verified that $q_\lambda(\pi; \mathcal{P}) \to q^*(\pi; \mathcal{P})$ as $\lambda \to 0$, which means optimal solutions can be identified by sampling from a constrained Gibbs distribution $q_\lambda$ with a sufficiently small $\lambda$.

As an alternative to directly sampling from the Gibbs distribution, constructing a parameterized distribution $p_\theta$ is often considered more efficient to sample a feasible route. This approach is adopted in variational annealing methods (Hibat-Allah et al., 2021; Sanokowski et al., 2023; Chen et al., 2023). We can construct an auto-regressive distribution that generates a route in a node-by-node manner. Given a problem instance $\mathcal{P}$, the policy for generating a solution $\pi$ of length $T$ can be decomposed as:

$$p_\theta(\pi; \mathcal{P}) = \prod_{t=1}^{T-1} p_\theta(\pi_{t+1}|\pi_{1:t}; \mathcal{P}), \tag{2}$$

where $p_\theta$ represents an auto-regressive neural network (Sutskever et al., 2014; Bahdanau et al., 2015; Vaswani et al., 2017) to predict the next element based on all preceding elements.

Let $P(x)$ and $Q(x)$ represent two probability distributions such that their supports satisfy $\text{supp}(P) \subseteq \text{supp}(Q)$. The Kullback-Leibler (KL) divergence from $Q$ to $P$ is defined as $\text{KL}(P||Q) :=$

$\mathbb{E}_{x \sim P(\cdot)} \left[ \log \frac{P(x)}{Q(x)} \right]$. In our framework, to reduce the discrepancy between the policy distribution $p_\theta$ and the Gibbs distribution $q_\lambda$, we minimize their KL divergence

$$\mathrm{KL}(p_\theta || q_\lambda) = \mathbb{E}_{p_\theta} [\log p_\theta] + \frac{1}{\lambda} \mathbb{E}_{p_\theta} [f(\pi; \mathcal{P})] + \log Z_\lambda - f^*(\mathcal{P}). \quad (3)$$

Then, eliminating $\theta$-independent terms $(\log Z_\lambda - f^*(\mathcal{P}))$ and $\lambda$-scaling in (3) yields the loss function

$$L(\theta; \mathcal{P}) := \mathbb{E}_{p_\theta(\cdot; \mathcal{P})}[f(\pi; \mathcal{P})] + \lambda \mathbb{E}_{p_\theta(\cdot; \mathcal{P})}[\log p_\theta(\pi; \mathcal{P})],$$

which contains the expectation of $f(\pi; \mathcal{P})$ over $p_\theta$ for concentration on lower function values and an entropy regularizer for encouraging exploration.

## 3 METHODOLOGY

### 3.1 CONSTRAINED AUTO-REGRESSIVE MODEL

Due to the complex constraints of the routing problem, infeasible solutions need to be excluded in the auto-regressive model. Transformer-based models for routing problems (Kool et al., 2018; Luo et al., 2023; Kwon et al., 2020) utilize masking techniques to avoid infeasible solutions. However, the effectiveness of these techniques relies on the routing problem's constraint structure. For problems with highly complex constraints, generating feasible solutions is challenging, as detailed in Appendix A.

To address this, we leverage the constraint structure of problem (1). Specifically, when constructing $p_\theta$, we must ensure that the probability of any solution $\pi$ violating the constraints is zero. This can be achieved by incorporating the indicator function $\mathbb{1}_C$ for the feasible set $C$. The conditional probability $p_\theta(\pi_{t+1}|\pi_{1:t}; \mathcal{P})$, represented by the neural network, should explicitly exclude infeasible actions. To formalize this, we introduce the potential set $S(\pi_{1:t})$, defined as:

$$S(\pi_{1:t}) := \{\pi_{t+1} : \exists \pi_{t+1:T} \in V^{T-t}, [\pi_{1:t}, \pi_{t+1:T}] \in C\},$$

which further induces the mask function $\mathbb{1}_{S(\pi_{1:t})}(\pi_{t+1})$. Based on this formulation, the conditional probability in (2) parameterized by the neural network takes the form:

$$p_\theta(\pi_{t+1}|\pi_{1:t}; \mathcal{P}) = \frac{e^{\phi_\theta(\pi_{t+1}|\pi_{1:t}; \mathcal{P})} \mathbb{1}_{S(\pi_{1:t})}(\pi_{t+1})}{\sum_{k=0}^{n} e^{\phi_\theta(k|\pi_{1:t}; \mathcal{P})} \mathbb{1}_{S(\pi_{1:t})}(k)},$$

where $\phi_\theta(\cdot|\pi_{1:t}; \mathcal{P})$ is produced by all intermediate layers. The auto-regressive model $p_\theta$ can produce feasible solutions step-by-step, forming the foundation for solving general routing problems.

### 3.2 LAZYMASK ALGORITHM

Most neural methods for routing problems involve handling constraints through masking mechanisms, which dynamically excludes actions that would lead to infeasible solutions during the construction. However, $S(\pi_{1:t})$ is sometimes computationally inaccessible for complex constraints since it may require an exhaustive lookahead until a complete solution is constructed. To circumvent this, we propose the LazyMask algorithm which works with an overestimation set $\hat{S}(\pi_{1:t})$ representing the currently known complementary set of actions that are deemed impossible. LazyMask adaptively establishes this set via lookahead initialization strategies and incrementally refining it through backtracking. The interaction between lookahead and backtracking enables efficient generation of feasible solutions under complex constraints. The detailed procedure is described in Algorithm 1.

**Backtrack.** LazyMask employs an adaptive backtracking mechanism to ensure solution feasibility while reducing unnecessary computation. At each step $t$, the algorithm examines the current overestimation $\hat{S}(\pi_{1:t})$. If this set is non-empty, the algorithm extends the partial route by sampling the next node $\pi_{t+1}$ according to the masked policy $p_\theta$, and then advances to step $t + 1$ to initialize the new overestimation set. Otherwise, if the set is empty, the algorithm's decision depends on the backtracking budget. If the budget is unreached, the algorithm backtracks to step $t - 1$ and refines $\hat{S}(\pi_{1:t-1})$ by removing the invalid node $\pi_t$. However, if the budget has been reached, it instead relaxes $\hat{S}(\pi_{1:t})$ to the set of all unvisited nodes to ensure a complete route can be generated.

For NP-hard combinatorial optimization problems with complex constraints, it is precisely due to the inherent complexity of the search space that generating a solution via a single forward pass is insufficient. The backtracking mechanism addresses this limitation by transforming it from a conventional one-pass forward model to a dynamic paradigm capable of both forward and backward operations. As a core role in our decoding algorithm, it significantly enhances the flexibility of route decoding and constitutes the most fundamental difference from previous neural solvers.

**Lookahead.** LazyMask offers great flexibility for initializing the overestimation set as long as $\hat{S}(\pi_{1:t})$ contains the potential set $S(\pi_{1:t})$. In this paper, we resort to lookahead strategies for this initialization. Single-step lookahead (SSL) is a common approach for ordinary routing problems that examines unvisited nodes for immediate constraint violations. Two-step lookahead (TSL) performs an additional step of lookahead to filter nodes that appear feasible under SSL but could lead to infeasible routes, which is adopted in (Bi et al., 2024) and termed as one-step PI masking. The specific implementations of these two strategies for TSPTW and TSPDL are provided in Appendix E.

---

**Algorithm 1** LazyMask algorithm

---

1: **Input:** routing problem instance $\mathcal{P}$, neural network $p_\theta$, backtracking budget $R$.
2: Initialize $\pi_1 := 0$, $t := 1$, $r := 0$ and the overestimation set $\hat{S}(\pi_1)$ either by SSL or TSL.
3: **while** $t \leq T - 1$ **do**
4:     **if** $\hat{S}(\pi_{1:t}) = \emptyset$ and $r \leq R$ **then**
5:         Update $\hat{S}(\pi_{1:t-1}) := \hat{S}(\pi_{1:t-1}) \backslash \{\pi_t\}$.
6:         Set $t := t - 1$, $r := r + 1$.
7:     **else**
8:         **if** $\hat{S}(\pi_{1:t}) = \emptyset$ **then**
9:             $\hat{S}(\pi_{1:t}) := V \backslash \{\pi_1, \ldots, \pi_t\}$.
10:        **end if**
11:        Calculate the probability $p_\theta(\cdot | \pi_{1:t}; \mathcal{P})$ using $\hat{S}(\pi_{1:t})$.
12:        Sample $\pi_{t+1} \sim p_\theta(\cdot | \pi_{1:t}; \mathcal{P})$.
13:        Set $t := t + 1$ and initialize $\hat{S}(\pi_{1:t})$ either by SSL or TSL.
14:     **end if**
15: **end while**
16: **Output:** route $\pi$.

---

### 3.3 REFINEMENT INTENSITY EMBEDDING

While our LazyMask algorithm is model-agnostic, standard dynamic features in existing autoregressive models (Kool et al., 2018; Kwon et al., 2020) are incompatible with backtracking. These features typically only reflect aggregated information from the partial route, such as the current time in TSPTW, implicitly assuming a one-pass forward construction. However, when backtracking occurs, this design renders the model state invariant to its search trace, leading to representation ambiguities that can hinder the model's learning ability. For instance, the model cannot distinguish whether the current partial route emerges from forward construction or backward correction.

To address this, we propose the refinement intensity embedding (RIE), designed to enrich the decoder's input with essential context about the search trace via the refinement intensity of overestimation sets. RIE is derived from two distinct refinement intensity features. The local feature quantifies the refinement count $c_t$ of the current $\hat{S}(\pi_{1:t})$, represented as a capped $N$-dimensional one-hot vector with its non-zero entry at index $\min(c_t + 1, N)$. The global feature signals whether the total number of backtracks has reached the budget $R$, encoded as a 2-dimensional one-hot vector. These features are concatenated and then projected to form the final RIE. The resulting RIE enhances the model's awareness of the search trace, resolving the representation ambiguities caused by backtracking.

### 3.4 TRAINING

For routing problems where identifying feasible solutions can be intractable (Savelsbergh, 1985), LazyMask with a large backtracking budget $R$ is inefficient during the early training stage. We therefore adopt a smaller $R$ during training to enhance computational efficiency. This practical choice, however, can lead to infeasible solutions, requiring a method to guide $p_\theta$ towards the feasible set. Thus,

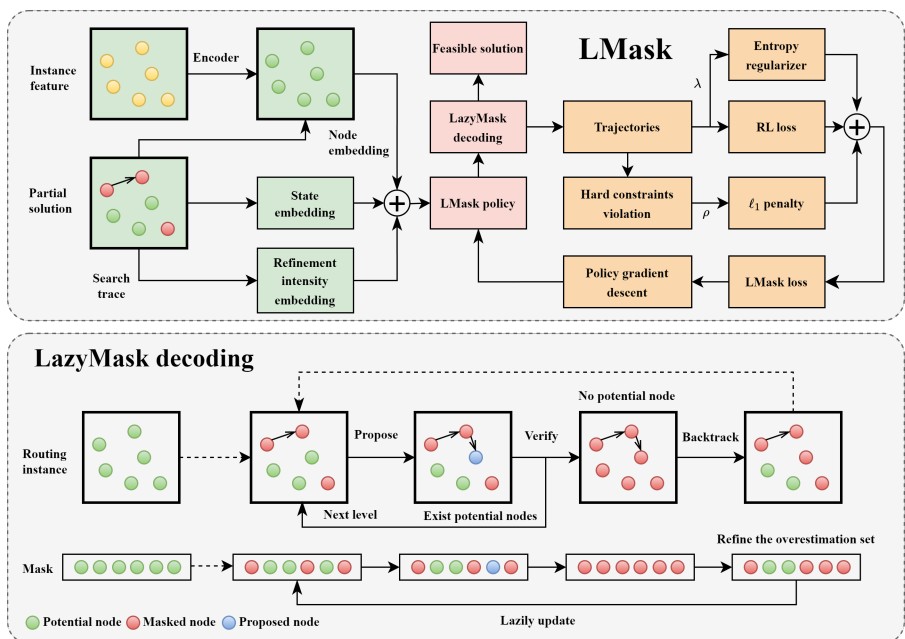

Figure 1: An illustrative overview of LMask: Up - the overall LMask framework. Down - the LazyMask decoding algorithm.

we employ the $\ell_1$ penalty function, a well-established technique in constrained optimization (Nocedal & Wright, 2006). This function specifically penalizes violations of complex constraints, such as time windows in TSPTW and draft limits in TSPDL, while simpler constraints like node visits are inherently satisfied by the design of $p_\theta$. The training objective is then formulated as

$$\min_\theta \quad \mathbb{E}_{\pi \sim p_\theta(\cdot;\mathcal{P})}\left[\Psi_\rho(\pi;\mathcal{P}) + \lambda \log p_\theta(\pi;\mathcal{P})\right],$$

where $\Psi_\rho(\pi;\mathcal{P}) := f(\pi;\mathcal{P}) + \rho \sum_{j=1}^{J} \max\left(c_j(\pi;\mathcal{P}), 0\right)$ is the $\ell_1$ penalty function, $\rho > 0$ is a given penalty parameter and $c_j(\pi;\mathcal{P})$ quantifies the violation of the $j$-th complex constraint. While previous studies (Tang et al., 2022; Ma et al., 2023) have explored $\ell_1$ penalty as soft constraints, we innovatively combine hard and soft constraints during training. It starts with hard constraints through backtracking and then turns to soft constraints for flexible optimization once the budget threshold is reached. The policy $p_\theta$ is trained using a standard policy gradient algorithm (Silver et al., 2014; Sutton & Barto, 2018) with further details provided in Appendix B. The overall LMask framework is illustrated in Figure 1.

## 4 THEORETICAL RESULTS

### 4.1 VALIDITY OF LAZYMASK ALGORITHM

To ensure the effectiveness of Algorithm 1 in solving constrained routing problems, we first prove that it always generates feasible solutions and has a non-zero probability of generating all feasible solutions. The following proposition shows the validity of the algorithm.

**Proposition 4.1.** *Suppose that the problem* (1) *is feasible, and that the backtracking budget in Algorithm 1 is set to $R = +\infty$. Then, i) any solution $\pi$ generated by Algorithm 1 is feasible; ii) Algorithm 1 assigns a non-zero probability to generate any feasible solution $\pi$.*

This proposition demonstrates that Algorithm 1 never generates infeasible solutions and no feasible solution is excluded. This ensures that the algorithm explores the entire feasible solution space with the distribution $p_\theta$, which acts as the foundation for further analysis of the algorithm's behavior in finding optimal solutions.

## 4.2 VALIDITY OF PROBABILISTIC MODEL

We further analyze the theoretical validity of the probabilistic model for the original routing problem by providing rigorous performance guarantees. Before delving into the formal theorem, it is essential to clarify a fundamental assumption supporting the analysis.

**Assumption 4.2.** We assume that the auto-regressive neural network $p_\theta$ has sufficient expressive power to approximate the target distribution $q_\lambda$. Specifically, the approximation error $\delta(\lambda)$ defined as follows satisfies:

$$\delta(\lambda) = \min_\theta \max_{\mathcal{P} \in \mathcal{D}} \mathrm{KL}(p_\theta(\cdot; \mathcal{P}) \parallel q_\lambda(\cdot; \mathcal{P})) \leq c/\lambda, \ \lambda > 0,$$

where $c$ is a small constant. Let $p_{\theta^*(\lambda)}$ be the corresponding optimal distribution.

Assumption 4.2 ensures that the auto-regressive neural network $p_\theta$ effectively parameterizes Gibbs distributions $q_\lambda$, with $\delta(\lambda)$ quantifying their approximation error. It is worth noting that as $\lambda$ is approaching zero, the target distribution $q_\lambda$ converges toward a point mass distribution, which is hard to approximate. Hence, we do not assume a uniform upper bound. We further give the following theorem to formalize the performance guarantees of the probabilistic model.

**Theorem 4.3.** *Suppose that Assumption 4.2 holds. We define $\Delta(\mathcal{P}) := \min_{\pi \in C \setminus \Pi^*} f(\pi; \mathcal{P}) - f^*(\mathcal{P})$. Then, for any $\epsilon > 0$ and $\Delta(\mathcal{P}) \geq \lambda > 0$, the following inequality holds:*

$$\mathbb{P}_{p_{\theta^*(\lambda)}}(f(\pi; \mathcal{P}) \geq f^*(\mathcal{P}) + \epsilon) \leq \frac{|C| \, \Delta(\mathcal{P}) e^{-\Delta(\mathcal{P})/\lambda}}{|\Pi^*| \max\{\epsilon, \Delta(\mathcal{P})\}} + \sqrt{\frac{c}{2\lambda}}.$$

Theorem 4.3 provides a probabilistic bound for the event that the solution $\pi \sim p_{\theta^*(\lambda)}(\cdot; \mathcal{P})$ is sampled with an objective value $f(\pi; \mathcal{P})$ significantly larger than the optimal value $f^*(\mathcal{P})$. The suboptimality gap $\Delta(\mathcal{P})$ serves as a measure of separation between optimal and suboptimal solutions in the feasible set $C$, while $\lambda$ controls the trade-off between exploration and concentration in the Gibbs distribution $q_\lambda$. A smaller $\lambda$ enhances concentration and suppresses suboptimal solutions more effectively but increases the approximation error $\sqrt{\frac{c}{2\lambda}}$, while a larger $\lambda$ reduces approximation error and improves computational feasibility but weakens the probability guarantee in the term of $e^{-\Delta(\mathcal{P})/\lambda}$. This trade-off of the entropy regularization coefficient $\lambda$ is intrinsic to the probabilistic model and underscores the interplay between concentration, exploration, and approximation quality.

## 5 EXPERIMENTS

We conduct experiments on two representative hard-constrained routing problems: TSPTW and TSPDL, using synthetic datasets of sizes $n = 50, 100$ across different hardness levels and a real-world TSPTW benchmark. All experiments are conducted on a server with NVIDIA Tesla A800 GPUs (80GB) and Intel Xeon Gold 6326 CPUs (256GB) at 2.90GHz. The data generation mechanism, implementation details, and additional results are available in Appendix E. The code of the proposed LMask framework is available at https://github.com/optsuite/LMask.

**Setup.** To evaluate the effectiveness of LMask, we compare it against several baselines. For *traditional solvers*, we adopt PyVRP (Wouda et al., 2024), LKH3 (Helsgaun, 2017) and OR-Tools (Furnon & Perron, 2024). For *neural solvers*, we consider PIP and PIP-D (Bi et al., 2024), which proactively mask actions that could lead to future infeasibility, and VSR-LKH (Zheng et al., 2023), which incorporates reinforcement learning into the local search process of LKH3. Additionally, we include two simple constructive heuristics with backtracking, Random-L and Random-C. We use greedy rollout with $8\times$ symmetric dihedral augmentation for all neural solvers as done in (Kwon et al., 2020), resulting in 8 solutions per instance. LMask adopts TSL as its default initialization strategy.

**Evaluation metrics.** We evaluate both the ability to handle complex constraints and the quality of feasible solutions. The instance infeasibility rate (Ins.) is the fraction of instances for which no feasible solution is found, and the solution infeasibility rate (Sol.) is the fraction of generated solutions that are infeasible. To assess solution quality, we report the average route length over the best feasible solutions, excluding instances for which no feasible solution is available. Additionally, the gap is computed with respect to PyVRP for TSPTW and LKH3 for TSPDL, and is averaged over instances where both the evaluated method and the reference solver find feasible solutions. Note that

the gap is computed on a different subset of instances from those used for the average route length, and serves as the primary metric for solution quality, as done in (Bi et al., 2024). Finally, we report the total inference time required to solve all instances in the dataset.

Table 1: Results on synthetic TSPTW datasets. Bold indicates the best among constructive methods.

| Nodes | Method | $n = 50$ | | | | | $n = 100$ | | | | |
|---|---|---|---|---|---|---|---|---|---|---|---|
| | | Infeasible Sol. | Inst. | Obj. | Gap | Time | Infeasible Sol. | Inst. | Obj. | Gap | Time |
| Easy | PyVRP | - | 0.00% | 7.31 | * | 1.7h | - | 0.00% | 10.19 | * | 4.3h |
| | LKH3 | - | 0.00% | 7.31 | 0.00% | 1.9h | - | 0.00% | 10.21 | 0.29% | 7.2h |
| | VSR-LKH | - | 0.00% | 7.31 | 0.00% | 4.3h | - | 0.00% | 10.19 | 0.08% | 16.4h |
| | OR-Tools | - | 0.00% | 7.32 | 0.21% | 1.7h | - | 0.00% | 10.33 | 1.43% | 4.3h |
| | Random-L | 56.02% | 0.04% | 14.55 | 99.23% | 1.3m | 95.17% | 9.37% | 30.66 | 201.20% | 6.1m |
| | Random-C | 62.31% | 0.03% | 19.12 | 162.03% | 1.4m | 98.17% | 27.30% | 44.20 | 333.74% | 6.1m |
| | PIP | 0.28% | 0.01% | 7.51 | 2.73% | 9s | 0.16% | **0.00%** | 10.57 | 3.78% | 29s |
| | PIP-D | 0.28% | **0.00%** | 7.50 | 2.60% | 10s | 0.05% | **0.00%** | 10.66 | 4.62% | 31s |
| | LMask | **0.06%** | **0.00%** | **7.45** | **2.02%** | 7s | **0.01%** | **0.00%** | **10.50** | **3.11%** | 17s |
| Medium | PyVRP | - | 0.00% | 13.03 | * | 1.7h | - | 0.00% | 18.72 | * | 4.3h |
| | LKH3 | - | 0.00% | 13.02 | 0.00% | 2.9h | - | 0.01% | 18.74 | 0.16% | 10.3h |
| | VSR-LKH | - | 0.00% | 13.03 | 0.01% | 8.2h | - | 0.00% | 18.72 | 0.00% | 8.7h |
| | OR-Tools | - | 15.12% | 13.01 | 0.12% | 1.5h | - | 0.52% | 18.98 | 1.40% | 4.3h |
| | Random-L | 98.31% | 32.18% | 18.91 | 47.04% | 1.6m | 100.00% | 100.00% | - | - | 5.8m |
| | Random-C | 91.17% | 8.18% | 21.04 | 61.91% | 1.6m | 100.00% | 100.00% | - | - | 5.9m |
| | PIP | 4.82% | 1.07% | 13.41 | 2.93% | 10s | 4.35% | 0.39% | 19.61 | 4.79% | 29s |
| | PIP-D | 4.14% | 0.90% | 13.46 | 3.31% | 9s | 3.46% | 0.03% | 19.80 | 5.76% | 31s |
| | LMask | **0.04%** | **0.00%** | **13.25** | **1.68%** | 6s | **0.05%** | **0.00%** | **19.51** | **4.23%** | 18s |
| Hard | PyVRP | - | 0.00% | 25.61 | * | 1.7h | - | 0.01% | 51.27 | * | 4.3h |
| | LKH3 | - | 0.52% | 25.61 | 0.00% | 2.3h | - | 0.95% | 51.27 | 0.00% | 1d8h |
| | VSR-LKH | - | 0.52% | 25.68 | 0.00% | 4.4h | - | 0.91% | 51.27 | 0.00% | 8.9h |
| | OR-Tools | - | 65.11% | 25.92 | 0.00% | 0.6h | - | 89.25% | 51.72 | 0.00% | 0.5h |
| | Random-L | 100.00% | 100.00% | - | - | 1.6m | 100.00% | 100.00% | - | - | 5.6m |
| | Random-C | 100.00% | 99.82% | 25.98 | 1.22% | 1.6m | 100.00% | 100.00% | - | - | 5.8m |
| | PIP | 5.65% | 2.85% | 25.73 | 0.18% | 9s | 31.74% | 16.68% | 51.48 | 0.37% | 28s |
| | PIP-D | 6.44% | 3.03% | 25.75 | 0.27% | 9s | 13.59% | 6.60% | 51.43 | 0.32% | 31s |
| | LMask | **0.00%** | **0.00%** | **25.71** | **0.10%** | 6s | **0.00%** | **0.00%** | **51.38** | **0.21%** | 18s |

Table 2: Results on synthetic TSPDL datasets. Bold indicates the best among constructive methods.

| Nodes | Method | $n = 50$ | | | | | $n = 100$ | | | | |
|---|---|---|---|---|---|---|---|---|---|---|---|
| | | Infeasible Sol. | Inst. | Obj. | Gap | Time | Infeasible Sol. | Inst. | Obj. | Gap | Time |
| Medium | LKH3 | - | 0.00% | 10.85 | * | 2.3h | - | 0.00% | 16.36 | * | 10.2h |
| | VSR-LKH | - | 0.00% | 10.85 | 0.08% | 3.8h | - | 0.00% | 16.35 | - 0.07% | 11.0h |
| | OR-Tools | - | 100.00% | - | - | 10.9s | - | 100.00% | - | - | 56.9s |
| | Random-L | 99.96% | 97.28% | 21.02 | 138.56% | 38s | 100.00% | 100.00% | - | - | 2.0m |
| | Random-C | 96.89% | 47.39% | 24.71 | 145.85% | 37s | 100.00% | 99.98% | 50.48 | 319.97% | 2.0m |
| | PIP | 1.75% | 0.17% | 11.23 | 3.59% | 8s | 2.50% | 0.16% | 17.68 | 8.10% | 21s |
| | PIP-D | 2.29% | 0.22% | 11.26 | 3.96% | 8s | 1.83% | 0.23% | 17.80 | 8.84% | 23s |
| | LMask | **0.03%** | **0.01%** | **11.14** | **2.75%** | 6s | **0.20%** | **0.05%** | **17.04** | **4.24%** | 15s |
| Hard | LKH3 | - | 0.00% | 13.25 | * | 2.6h | 0.00% | 0.00% | 20.76 | * | 15.8h |
| | VSR-LKH | - | 0.00% | 13.25 | 0.05% | 6.0h | - | 0.00% | 20.75 | - 0.05% | 17.2h |
| | OR-Tools | - | 100.00% | - | - | 10.6s | - | 100.00% | - | - | 56.8s |
| | Random-L | 100.00% | 99.96% | 22.2 | 132.40% | 37s | 100.00% | 100.00% | - | - | 2.0m |
| | Random-C | 99.90% | 94.05% | 25.55 | 135.68% | 37s | 100.00% | 100.00% | - | - | 2.0m |
| | PIP | 4.83% | 2.39% | 13.63 | 3.42% | 8s | 29.34% | 21.65% | 22.35 | 12.87% | 20s |
| | PIP-D | 4.16% | 0.82% | 13.79 | 4.28% | 8s | 13.51% | 8.43% | 22.90 | 12.53% | 23s |
| | LMask | **0.19%** | **0.04%** | **13.57** | **2.52%** | 6s | **0.80%** | **0.26%** | **21.63** | **4.34%** | 15s |

## 5.1 PERFORMANCE ON SYNTHETIC DATASETS

The results on synthetic TSPTW and TSPDL datasets are presented in Tables 1 and 2. On TSPTW datasets, LMask consistently achieves near-zero infeasibility rates, significantly surpassing OR-Tools,

PIP, PIP-D and heuristics across different hardness levels. Compared to the traditional solvers PyVRP and LKH3, LMask is substantially efficient due to fast network inference. Although VSR-LKH is also learning based, it is implemented on top of the computationally intensive LKH3 pipeline and therefore inherits its runtime profile, remaining orders of magnitude slower than LMask. Regarding solution quality, we observe that while PIP-D can improve feasibility over PIP, this sometimes comes at the cost of larger gaps. In contrast, LMask delivers improved feasibility and lower optimality gaps than PIP and PIP-D across all settings, while maintaining competitive inference times. The advantages of LMask become even more pronounced on TSPDL datasets, showing significant improvements in both feasibility and solution quality. These results highlight that LMask can effectively generate higher-quality solutions while significantly reducing the occurrence of infeasible solutions.

## 5.2 GENERALIZATION AND SCALABILITY

In Table 3, we further assess the generalization ability of LMask on the well-known TSPTW benchmark (Dumas et al., 1995). Across all problem sizes, LMask surpasses neural baselines in both solution feasibility and quality. Notably, as the problem size increases, the performance of PIP and PIP-D degrades significantly, whereas LMask remains robust, demonstrating strong generalization across problem sizes. Additional experiments with varying maximum time window widths in Appendix E further confirm the generalization ability of LMask. Beyond generalization, LMask also exhibits notable scalability. Results on hard TSPTW-500 in Appendix E validate its scalability.

Table 3: Results on the TSPTW benchmark.

| Nodes | $n = 20$ | | | $n = 40$ | | | $n = 60$ | | | $n = 80$ | | |
|---|---|---|---|---|---|---|---|---|---|---|---|---|
| Method | Infeas. | Obj. | Gap | Infeas. | Obj. | Gap | Infeas. | Obj. | Gap | Infeas. | Obj. | Gap |
| PIP | 5.0% | 337.00 | 5.2% | 45.0% | 428.09 | 4.6% | 20.0% | 580.25 | 11.5% | 22.2% | 644.43 | 8.7% |
| PIP-D | 5.0% | 336.63 | 5.2% | 25.0% | 460.27 | 6.3% | 40.0% | 608.67 | 13.1% | 66.7% | 662.67 | 12.0% |
| LMask | 0.0% | 326.55 | 0.7% | 0.0% | 445.50 | 1.0% | 0.0% | 530.60 | 4.3% | 0.0% | 615.11 | 3.5% |

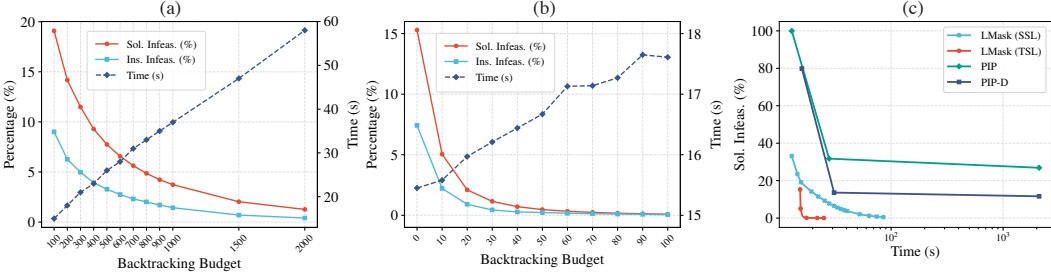

Figure 2: Empirical evaluation of the backtracking mechanism on hard TSPTW-100. (a)-(b) Impact of the backtracking budget $R$ under SSL and TSL initializations, respectively. (c) Solution infeasibility versus inference time for LMask with different overestimation initial strategies and baseline methods.

## 5.3 EMPIRICAL ANALYSIS ON BACKTRACKING

**Effect of the backtracking budget.** Here we investigate how the backtracking budget $R$ influences the performance of LMask under both SSL and TSL initialization strategies on the hard TSPTW-100 dataset. As shown in Figure 2 (a)(b), for both strategies, inference time exhibits a nearly linear growth with respect to $R$, with an increase of approximately 2 seconds per 100 additional backtracking budget, demonstrating manageable computational overhead. On the other hand, infeasibility rates decrease sharply at small values of $R$, indicating substantial early-stage gains in feasibility. These results highlight that larger backtracking budgets substantially improve solution feasibility with modest increases in runtime.

**Backtracking vs. lookahead.** The reduction in solution infeasibility as the inference budget increases for different methods is illustrated in Figure 2 (c). The inference budget is scaled by increasing the backtracking budget $R$ for LMask, whereas for PIP and PIP-D, it is adjusted by increasing the lookahead depth from one to three. With TSL, LMask attains a zero solution infeasibility rate within

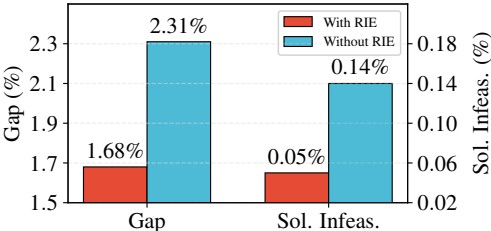

Figure 3: Effect of RIE.

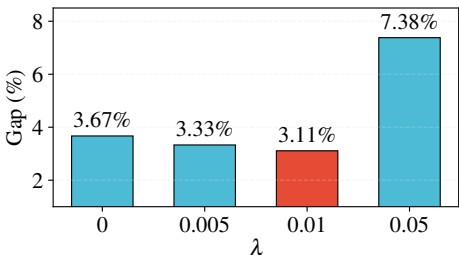

Figure 4: Effect of entropy term.

30 seconds. Even under the less accurate SSL, LMask drives the infeasibility rate down to the second lowest level by allocating a larger backtracking budget. However, PIP and PIP-D exhibit unacceptably high infeasibility rates under SSL. Increasing the lookahead step from 2 to 3 induces an order of magnitude rise in inference time while yielding only marginal gains and the outcomes remain inferior to LMask under SSL with significantly less inference time. These results demonstrate that backtracking combined with a lightweight lookahead initialization is more efficient than methods that rely exclusively on deeper lookaheads.

**Additional results.** To thoroughly analyze the interaction of backtracking and the overestimation set initialization strategy, we present results on medium TSPTW-50 for all 16 combinations of their usage at training and inference in Figure 5 of Appendix E.3. We also visualize the decoding process of LMask in Figures 6 and 7 of Appendix E.4, which provides an in-depth analysis of the circumstances under which backtracking is efficient and those in which it is not.

### 5.4 ABLATION STUDY

**Refinement intensity embedding.** In Figure 3, we present the results on medium TSPTW-50 with and without RIE. The results confirm that the inclusion of RIE leads to a substantial reduction in optimality gap. Concurrently, it also fosters a tangible improvement in solution feasibility.

**Entropy term.** In Figure 4, we report the results on easy TSPTW-100 for models trained with different entropy coefficients $\lambda$. The optimality gap exhibits a non-monotonic pattern. It decreases as $\lambda$ increases from 0 to 0.01, achieving the best performance at $\lambda = 0.01$, but then rises significantly at $\lambda = 0.05$. This reflects the intrinsic trade-off between exploration and concentration in the probabilistic model, and suggests that choosing an appropriate entropy coefficient can improve solution optimality. This observation also aligns with our theoretical analysis in Section 4.2.

## 6 CONCLUSION

In this paper, we propose a novel framework, LMask, for solving hard-constrained routing problems by introducing innovative masking mechanisms. By addressing feasibility through lazy masking, our approach can generate feasible solutions efficiently through the transformer-based model with RIE. We provide theoretical guarantees demonstrating that our approach preserves both feasibility and optimality. Extensive experiments on TSPTW and TSPDL reveal that LMask achieves state-of-the-art feasibility rates and solution quality. Although our current framework is only applied to limited problem types, future work may explore its extension to more general combinatorial optimization problem with simultaneously reduced infeasibility and the solution gap.

## 7 ACKNOWLEDGMENTS

The computational resources were supported by the Center for Intelligent Computing and Song-Shan Lake HPC Center (SSL-HPC) in Great Bay University, Dongguan, China. This research was supported in part by National Key Research and Development Program of China under the grant number 2024YFA1012900, the National Natural Science Foundation of China under the grant numbers 12331010 and 12288101, and the Natural Science Foundation of Beijing, China under the grant number Z230002. We also thank the anonymous reviewers for their valuable feedback.

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

# LMask: Learn to Solve Constrained Routing Problems with Lazy Masking (Appendix)

## A COMBINATORIAL OPTIMIZATION FORMULATION OF ROUTING PROBLEMS

### A.1 TSPTW AND TSPDL FORMULATION

The traveling salesman problem with time windows (TSPTW) is a well-known combinatorial optimization problem that extends the classic traveling salesman problem (TSP) by introducing additional time window constraints. The objective of TSPTW is to find the shortest route for a salesman, starting and ending at a designated depot, and visiting a given set of customers exactly once. Let node $0$ represent the depot and $V_c := \{1, 2, \ldots, n\}$ represent the set of customer nodes. Each customer $i \in V_c$ has an associated time window $[e_i, l_i]$ during which it must be served. Early arrivals are allowed, meaning that the salesman can arrive at node $i$ before the ready time $e_i$. However, in this case the salesman has to wait until service can begin at node $i$. This problem can be formulated as:

$$
\begin{aligned}
\min \quad & f(\pi; \mathcal{P}) = \sum_{i=1}^{n} \left\| x_{\pi_i} - x_{\pi_{i+1}} \right\| + \left\| x_{\pi_{n+1}} - x_{\pi_1} \right\| \\
\text{s.t.} \quad & c_i(\pi; \mathcal{P}) = \tau_{i+1} - l_{\pi_{i+1}} \leq 0, \quad i = 1, \ldots, n, \\
& d_i(\pi; \mathcal{P}) = \sum_{t=1}^{n+1} \mathbb{1}_{\pi_t = i} - 1 = 0, \quad i = 1, \ldots, n,
\end{aligned}
\tag{4}
$$

where $\tau_{i+1}$ represents the time to begin service at node $\pi_{i+1}$ in route $\pi$. This time is derived recursively: $\tau_{i+1} = \max\left(\tau_i + w_{\pi_i, \pi_{i+1}}, e_{\pi_{i+1}}\right)$, $i = 1, \ldots, n$, with $\tau_1 = 0$. Here $w_{\pi_i, \pi_{i+1}}$ is the total duration from $\pi_i$ to $\pi_{i+1}$, which includes both the service time at $\pi_i$ and the travel time from $\pi_i$ to $\pi_{i+1}$.

The traveling salesman problem with draft limits (TSPDL) frequently arises in marine transportation scenarios, where the load-carrying limits of vessels must be respected. In this problem, node $0$ serves as the depot, and $V_c := \{1, 2, \ldots, n\}$ represents the set of port nodes. Each port $i \in V_c$ is associated with a demand $q_i > 0$ and a draft limit $D_i$. The draft limit $D_i$ at port $i$ specifies the maximum cumulative load a vessel can carry after visiting that port. The depot at node $0$ is assumed to have zero demand, meaning $q_0 = 0$. This problem can be formulated as:

$$
\begin{aligned}
\min \quad & f(\pi; \mathcal{P}) = \sum_{i=1}^{n} \left\| x_{\pi_i} - x_{\pi_{i+1}} \right\| + \left\| x_{\pi_{n+1}} - x_{\pi_1} \right\| \\
\text{s.t.} \quad & c_i(\pi; \mathcal{P}) = \delta_{i+1} - D_{\pi_{i+1}} \leq 0, \quad i = 1, \ldots, n, \\
& d_i(\pi; \mathcal{P}) = \sum_{t=1}^{n+1} \mathbb{1}_{\pi_t = i} - 1 = 0, \quad i = 1, \ldots, n,
\end{aligned}
\tag{5}
$$

where $\delta_{i+1}$ denotes the accumulated load after visiting node $\pi_{i+1}$ in route $\pi$. The accumulated load can be calculated as $\delta_{i+1} = \sum_{t=1}^{i+1} q_{\pi_t}$.

### A.2 FEASIBILITY DILEMMA IN TSPTW

Bi et al. (2024) point out that the core of feasibility masking in neural constructive solvers is to filter out invalid actions that violate constraints, based on the assumption that the global feasibility can be decomposed into the feasibility of each node selection step, and that ground truth masks are obtainable for each step. We describe this issue more formally using the notation defined in Section 3.1. For simple constraints in other routing problems, there always exists a feasible action for each step. For example, at each step in CVRP, it is possible to return to the depot to perform an

action that satisfies the capacity constraints. This indicates that the potential set $S$ can be precisely determined, enabling an efficient masking technique to ensure the feasibility. Since $S \neq \emptyset$ at each step, a valid action is always available. This characteristic ensures that the solution space remains connected, avoiding situations where no valid action can be performed due to capacity violations.

However, not all constraints can precisely determine the potential set $S$, which may even be empty at certain decision steps. In TSPTW, nodes are masked out if they have been visited or cannot be visited before their time windows close. The feasibility of selecting a node at a particular step impacts the current time, thereby affecting all subsequent selections due to the interdependence imposed by time windows constraints. Therefore, focusing solely on the local feasibility does not guarantee overall feasibility. Once a node is selected, the decision becomes irreversible, potentially leading to infeasible situations after several steps. Nevertheless, it is impractical to compute full global feasibility to obtain the exact potential set $S$, which is typically considered an NP-hard problem. There is a dilemma between solution feasibility and computational costs under these complex constraints.

## B    TRAINING DETAILS

In this section, we elaborate on the formulation of the penalty function $\Psi_\rho(\pi; \mathcal{P})$ and the policy gradient algorithm used for training $p_\theta$.

### B.1    PENALTY FUNCTION FORMULATION

The penalty function $\Psi_\rho(\pi; \mathcal{P})$ is designed to guide the policy towards feasible solutions by penalizing constraint violations. It combines the primary objective function $f(\pi; \mathcal{P})$ with terms representing the severity of constraint violations.

For TSPTW, given a complete route $\pi = (\pi_1 = 0, \pi_2, \ldots, \pi_T)$, let $\tau_t$ denote the time to begin service at node $\pi_t$ in the route $\pi$ and $l_{\pi_t}$ denote the due time of $\pi_t$, for $t = 1, 2, \ldots, T$. The penalty function is defined as

$$\Psi_\rho(\pi; \mathcal{P}) = f(\pi; \mathcal{P}) + \rho \sum_{t=1}^{T} \max(\tau_t - l_{\pi_t}, 0).$$

Here, the term $\sum_{t=1}^{T} \max(\tau_t - l_{\pi_t}, 0)$ quantifies the total amount by which time windows are violated.

Similarly, for TSPDL, we consider a route $\pi = (\pi_1 = 0, \pi_2, \ldots, \pi_T)$. Let $\delta_t$ represent the accumulated load after visiting node $\pi_t$ and $D_{\pi_t}$ denote the draft limit of node $\pi_t$, for $t = 1, 2, \ldots, T$. The penalty function for TSPDL is formulated as

$$\Psi_\rho(\pi; \mathcal{P}) = f(\pi; \mathcal{P}) + \rho \sum_{t=1}^{T} \max(\delta_t - D_{\pi_t}, 0).$$

Here $\sum_{t=1}^{T} \max(\delta_t - D_{\pi_t}, 0)$ represents the total excess load beyond the specified draft limits.

### B.2    TRAINING ALGORITHM

The training objective, as introduced in the main text, is formulated as

$$\min \quad \mathbb{E}_{\pi \sim p_\theta(\cdot; \mathcal{P})} \left[ \Psi_\rho(\pi; \mathcal{P}) + \lambda \log p_\theta(\pi; \mathcal{P}) \right], \tag{6}$$

which is expressed in the form of an expectation. Hence we can minimize it by various stochastic policy gradient methods. It is well known that the policy gradient of (6) is given by

$$\mathbb{E}_{\pi \sim p_\theta(\cdot; \mathcal{P})} \left[ A(\pi; \mathcal{P}) \nabla_\theta \log p_\theta(\pi; \mathcal{P}) \right],$$

where $A(\pi; \mathcal{P}) := \Psi_\rho(\pi; \mathcal{P}) - b(\mathcal{P}) + \lambda \log p_\theta(\pi; \mathcal{P})$ denotes the advantage function and $b(\mathcal{P})$ is a constant with respect to the parameter $\theta$, referred to the baseline. We then utilize the Monte Carlo method to estimate the policy gradient. Specifically, we sample $N$ routes $\{\pi^j\}_{j=1}^{N}$ and then estimate the expectation with sample average

$$\hat{g}(\theta; \mathcal{P}) = \frac{1}{N} \sum_{j=1}^{N} A(\pi^j; \mathcal{P}) \nabla_\theta \log p_\theta(\pi^j; \mathcal{P}).$$

In routing problems, empirically, a shared baseline has the desirable effect of reducing variance in the sample estimate for the policy gradient, as demonstrated in Kwon et al. (2020). This shared baseline writes $b(\mathcal{P}) = \frac{1}{N} \sum_{j=1}^{N} \Psi_\rho(\pi^j; \mathcal{P})$.

So far, we have described how to train the policy $p_\theta$ on a given instance $\mathcal{P}$. The training can be extended to a dataset of instances, as shown in Algorithm 2.

---

**Algorithm 2** LMask training procedure

---

**Input:** data distribution $\mathcal{D}$, neural network $p_\theta$, number of training steps $K$, backtracking budget $R$.
**for** $k = 1, \ldots, K$ **do**
  Sample a batch of instances $\{\mathcal{P}_i\}_{i=1}^{B}$ from the data distribution $\mathcal{D}$.
  Employ the LazyMask algorithm with backtracking budget $R$ to sample $N$ routes $\{\pi^{ij}\}_{j=1}^{N}$ from $p_{\theta^k}$ for each instance $\mathcal{P}_i$, where $i = 1, \ldots, B$.
  Compute the stochastic gradient:

$$\hat{g}(\theta^k) = \frac{1}{BN} \sum_{i=1}^{B} \sum_{j=1}^{N} A(\pi^{ij}; \mathcal{P}_i) \nabla_\theta \log p_\theta(\pi^{ij}; \mathcal{P}_i) \mid_{\theta=\theta^k}.$$

  Update $\theta^k$ using $\hat{g}(\theta^k)$ with SGD or ADAM optimizer.
**end for**

---

## C    MODEL ARCHITECTURE

LMask adopts an encoder-decoder architecture inherited from POMO (Kwon et al., 2020). The encoder transforms static features of a problem instance into node embeddings through self-attention mechanism. Based on node embeddings, dynamic features of the current partial route and the refinement intensity embedding, the decoder then auto-regressively generates the conditional probability through the cross-attention mechanism.

### C.1    MULTI-HEAD ATTENTION

An attention function takes a set of queries and a separate set of keys and values as input, and outputs a weighted sum of values for each query. Self-attention means that the query set and the key-value set come from the same sequence, while cross-attention means the query set and the key-value set comes from two different sequences. These queries, keys and values are packed into matrices $Q \in \mathbb{R}^{n_1 \times d}$, $K \in \mathbb{R}^{n_2 \times d}$, $V \in \mathbb{R}^{n_2 \times d}$ for implementation efficiency, where $n_1$ is the number of queries, $n_2$ is the number of key-value pairs and $d$ is the hidden dimension. Firstly, attention weights are computed by scaled dot-product between the queries and keys, followed by a softmax function

$$A = \text{Softmax}\left(\frac{QK^{\mathrm{T}}}{\sqrt{d}} + M\right),$$

where the softmax function should be understood in a row-wise manner with $\text{Softmax}(\boldsymbol{x})_i := \frac{\exp(x_i)}{\sum_{j=1}^{N} \exp(x_j)}$ for an $N$ dimensional vector $\boldsymbol{x}$, and $M$ is an optional attention mask that prevents attending to certain positions, which can be done by setting elements to $-\infty$. Then, the attention output is calculated as the sum of the values, weighted by the attention weights: $Z = AV$. The whole attention function is defined by

$$\text{Attention}(Q, K, V; M) = \text{Softmax}\left(\frac{QK^{\mathrm{T}}}{\sqrt{d}} + M\right) V.$$

Multi-head attention further takes information from different representation subspaces into consideration. It starts by linearly projecting the queries, keys, and values onto $H$ subspaces. Subsequently, the attention function is performed on each subspace in parallel. Lastly, these attention outputs are concatenated and projected back to the original embedding space. In summary, the multi-head attention operation can be formulated as

$$\text{MHA}(Q, K, V; M) = [Z_1, Z_2, \ldots, Z_H] W^O, \tag{7}$$

where

$$Z_i = \text{Attention}(QW_i^Q, KW_i^K, VW_i^V; M), \quad i = 1, \ldots, H,$$

and $W^O$, $W_i^Q$, $W_i^K$, $W_i^V$ are learnable projection matrices. The general formulation includes the mask $M$ to support masked attention scenarios. In cases where no masking is applied (i.e., $M$ is effectively a zero matrix or conceptually omitted), the operation is often simplified as $\text{MHA}(Q, K, V)$.

## C.2 ENCODER

The encoder produces embeddings of all input nodes. Static features of the problem instance are first projected to the embedding space to obtain initial embeddings $\boldsymbol{h}^{(0)} \in \mathbb{R}^{(n+1) \times d}$. These static features vary depending on the specific problem considered. For TSPTW, they include node coordinates and time windows, while for TSPDL, they include node coordinates, demands, and draft limits. Then the embeddings are updated through a stack of $L$ attention layers, each consisting of two sublayers, one multi-head attention sublayer and one feed-forward sublayer. Furthermore, residual connections and instance normalization are employed. A single attention layer can be formulated as

$$\hat{\boldsymbol{h}}^\ell = \text{IN}\left(\boldsymbol{h}^\ell + \text{MHA}^\ell(\boldsymbol{h}^\ell, \boldsymbol{h}^\ell, \boldsymbol{h}^\ell)\right), \quad \ell = 0, \ldots, L-1,$$

$$\boldsymbol{h}^{\ell+1} = \text{IN}\left(\hat{\boldsymbol{h}}^\ell + \text{FFN}^\ell(\hat{\boldsymbol{h}}^\ell)\right), \quad \ell = 0, \ldots, L-1,$$

where IN represents the instance normalization, MHA is the multi-head attention, as given by (7), and the fully-connected feed forward network $\text{FFN}^\ell$ is applied in a row-wise manner with $\text{FFN}^\ell(\boldsymbol{x}) = \max\left(\boldsymbol{x}W_1^\ell + b_1^\ell, 0\right)W_2^\ell + b_2^\ell$ for a row vector $\boldsymbol{x}$.

## C.3 DECODER

The decoder auto-regressively generates the conditional probability over available nodes based on the node embeddings and the current partial route. At decision step $t$, $1 \le t \le T-1$, a partial route $\pi_{1:t}$ is assumed to have been constructed.

Previous works (Kwon et al., 2020; Berto et al., 2024) begin by constructing a query through a projection of the current node embedding $h_{\pi_t}$ and dynamic features $s_t$ into the embedding space: $q_t = h_{\pi_t}W_c^1 + s_t W_c^2$. This is represented as the sum of the node embedding and the state embedding. The dynamic features $s_t$ typically aggregate information accumulated up to the current decision step $t$. For instance, in TSPTW, $s_t$ represents the current time, and in TSPDL, $s_t$ represents the current load. However, in the presence of backtracking, relying solely on such features can lead to representation ambiguities. The model state, defined only by $h_{\pi_t}$ and $s_t$, becomes invariant to the specific search trace, potentially hindering the model's ability to learn meaningful distinctions.

To address this limitation, we introduce the refinement intensity embedding (RIE). The RIE enhances the model's awareness of the search trace by explicitly encoding information about refinement intensity. The refinement intensity is captured by a set of dedicated features, denoted by $\zeta_t$, which are subsequently projected to form the RIE $\zeta_t W_c^3$. $\zeta_t$ is conceptualized from two distinct perspectives. First, the local feature, represented by $\varphi_t$, quantifies $c_t$, the number of refinements applied to the overestimation set $\hat{S}(\pi_{1:t})$ corresponding the current partial route $\pi_{1:t}$. This count $c_t$ is transformed into a capped one-hot vector $\varphi_t \in \{0, 1\}^N$ of predefined length $N$: if $c_t < N$, the $(c_t + 1)$-th element of $\varphi_t$ is 1; otherwise, the $N$-th element is 1, with all other elements being 0. Second, the global refinement status, captured by $\xi_t$, is a binary feature indicating whether $u_t$, the total backtracking count accumulated during the decoding process, has reached the backtracking budget $R$. This is defined as $\xi_t = [1, 0]$ if $u_t < R$, and $\xi_t = [0, 1]$ if $u_t \ge R$. These two types of features are concatenated to yield the overall refinement intensity features $\zeta_t = [\varphi_t, \xi_t] \in \mathbb{R}^{N+2}$. The query $q_t$ is then constructed by incorporating the RIE: $q_t = h_{\pi_t}W_c^1 + s_t W_c^2 + \zeta_t W_c^3$.

To ensure that the subsequent attention mechanism and probability calculation only consider valid next nodes, we define a mask $M_t \in \{0, -\infty\}^{n+1}$. This mask vector is derived from $\hat{S}(\pi_{1:t})$ such that $M_t(i) = 0$ if node $i$ belongs to $\hat{S}(\pi_{1:t})$, and $M_t(i) = -\infty$ otherwise. Using this mask, a context embedding, $h_t^c$, is subsequently obtained via a masked multi-head cross-attention layer. This layer

employs $q_t$ as the query, with static node embeddings $\boldsymbol{h}^L$ projected by $W_g^K$ and $W_g^V$ to serve as keys and values respectively, and incorporating the mask $M_t$:

$$h_t^c = \text{MHA}(q_t, \boldsymbol{h}^L W_g^K, \boldsymbol{h}^L W_g^V; M_t).$$

Next, the logits are computed with a single attention head:

$$z = \frac{h_t^c (\boldsymbol{h}^L W^K)^{\text{T}}}{\sqrt{d}}.$$

Finally, the logits $z$ are transformed into a conditional probability distribution over available nodes

$$p_\theta(\cdot \mid \pi_{1:t}; \mathcal{P}) = \text{Softmax}(C \tanh(z + M_t)),$$

where $C > 0$ is a clipping constant for the $\tanh$ function.

# D    PROOF OF THEORETICAL RESULTS

## D.1    PROOF OF PROPOSITION 4.1

For clarity, we restate Proposition 4.1 here before providing the proof.

**Proposition 4.1.** *Suppose that problem (1) is feasible, and that the backtracking budget in Algorithm 1 is set to $R = +\infty$. Then, i) any solution $\pi$ generated by Algorithm 1 is feasible; ii) Algorithm 1 assigns a non-zero probability to generate any feasible solution $\pi$.*

*Proof.* i) Let the route $\pi$ be generated by Algorithm 1. Since $\hat{S}(\pi_{1:T-1})$ only needs to validate the feasibility of the last node $\pi_T$, the estimation $\hat{S}(\pi_{1:T-1})$ is exact, that is, $S(\pi_{1:T-1}) = \hat{S}(\pi_{1:T-1})$. It follows that $\pi_T \in S(\pi_{1:T-1})$. By the definition of $S(\pi_{1:T-1})$, this implies that $\pi$ is a feasible route.

ii) Let $\pi$ be a feasible route. For $t = 1, \ldots, T-1$, the definition of $S(\pi_{1:t})$ yields $\pi_{t+1} \in S(\pi_{1:t})$. Since $S(\pi_{1:t}) \subseteq \hat{S}(\pi_{1:t})$, it follows that $\pi_{t+1} \in \hat{S}(\pi_{1:t}) \neq \emptyset$ for $t = 1, \ldots, T-1$. Therefore, the Algorithm 1 assigns a probability $p_\theta(\pi_{t+1}|\pi_{1:t}; \mathcal{P}) > 0$ to generate $\pi_{t+1}$. The total probability of generating the complete route $\pi$ is strictly positive, i.e., $p_\theta(\pi) = \prod_{t=1}^{T-1} p_\theta(\pi_{t+1}|\pi_{1:t}; \mathcal{P}) > 0$. Thus, route $\pi$ can be generated by Algorithm 1.

This completes the proof. $\qquad\qquad\qquad\qquad\qquad\qquad\qquad\qquad\qquad\qquad\qquad\qquad\square$

## D.2    PROOF OF THEOREM 4.3

For clarity, we restate Theorem 4.3 here before providing the proof.

**Theorem 4.3.** *Suppose that Assumption 4.2 holds. We define $\Delta(\mathcal{P}) := \min_{\pi \in C \backslash \Pi^*} f(\pi; \mathcal{P}) - f^*(\mathcal{P})$. Then, for any $\epsilon > 0$ and $\Delta(\mathcal{P}) \geq \lambda > 0$, the following inequality holds:*

$$\mathbb{P}_{p_{\theta^*(\lambda)}}(f(\pi; \mathcal{P}) \geq f^*(\mathcal{P}) + \epsilon) \leq \frac{|C| \, \Delta(\mathcal{P}) e^{-\Delta(\mathcal{P})/\lambda}}{|\Pi^*| \max\{\epsilon, \Delta(\mathcal{P})\}} + \sqrt{\frac{c}{2\lambda}}.$$

*Proof.* For convenience, we omit all notations $\mathcal{P}$ in this proof. Let $\pi$ denote a random variable drawn from the distribution $p_{\theta^*(\lambda)}$, and define the event $A := \{f(\pi) \geq f^* + \epsilon\}$. We aim to bound the probability of the event $A$. To this end, let us first bound the difference in probabilities of the $A$ under the distribution $p_{\theta^*(\lambda)}$ and $q_\lambda$ using the KL divergence. Then, we apply Markov's inequality to derive a tail bound on the event probability under $q_\lambda$. Finally, using properties of the Gibbs distribution, we derive a convergence bound that characterizes how this probability behaves as $\lambda$ decreases.

1) We begin by introducing Pinsker's inequality (Pinsker, 1964), which bounds the total variation (TV) distance in terms of the KL divergence. Given two distributions $P$ and $Q$ over a finite domain $D$, the TV distance is defined as

$$\text{TV}(P, Q) := \max_{E \subseteq D} |P(E) - Q(E)|.$$

where $E \subseteq D$ represents any measurable event of the domain $D$. Pinsker's inequality states that the TV distance and KL divergence satisfy the following inequality:

$$\text{TV}(P, Q) \leq \sqrt{\frac{1}{2} \text{KL}(P \parallel Q)}.$$

Using Pinsker's inequality, we can bound the probability difference for the given event $A$ under two given distributions $p_{\theta^*(\lambda)}$ and $q_\lambda$:

$$|p_{\theta^*(\lambda)}(A) - q_\lambda(A)| \leq \text{TV}(p_{\theta^*(\lambda)}, q_\lambda) \leq \sqrt{\frac{1}{2} \text{KL}(p_{\theta^*(\lambda)} \| q_\lambda)} \leq \sqrt{\frac{\delta(\lambda)}{2}} \leq \sqrt{\frac{c}{2\lambda}}, \quad (8)$$

where the first inequality follows from the definition of the TV distance, the second uses Pinsker's inequality and the last is based on the Assumption 4.2. This result provides a way to control the discrepancy in the probability of event $A$ under the two distributions $p_{\theta^*(\lambda)}$ and $q_\lambda$, in terms of the approximation error $\delta(\lambda)$.

2) Then, we analyze the tail probability of $q_\lambda$ using Markov's inequality. If $X$ is a nonnegative random variable and $\epsilon > 0$, then the probability that $X$ is at least a is at most the expectation of $X$ divided by $\epsilon$:

$$\mathbb{P}(X \geq a) \leq \frac{\mathbb{E}X}{\epsilon}.$$

Using Markov's inequality, we can give the following tail bound:

$$\mathbb{P}_{q_\lambda}(f(\pi) \geq f^* + \epsilon) \leq \frac{\mathbb{E}_{q_\lambda}[f(\pi) - f^*]}{\epsilon}. \quad (9)$$

As $f$ is defined over a discrete domain, it has finitely many function values and suboptimality gap $\Delta > 0$. It can be observed that $q_\lambda(f(\pi) \geq f^* + \epsilon) = q_\lambda(f(\pi) \geq f^* + \Delta)$ when $0 < \epsilon \leq \Delta$. We can conclude the following inequality from (9):

$$\mathbb{P}_{q_\lambda}(f(\pi) \geq f^* + \epsilon) \leq \frac{\mathbb{E}_{q_\lambda}[f(\pi) - f^*]}{\max\{\epsilon, \Delta\}}. \quad (10)$$

3) Since $q_\lambda$ converges to the target distribution $q^*$ as $\lambda \to 0$, we analyze the asymptotic behavior of the expected suboptimality gap $\mathbb{E}_{q_\lambda}[f(\pi) - f^*]$ as $\lambda$ decreases. Specifically, we write the expectation:

$$\mathbb{E}_{q_\lambda}[f(\pi) - f^*] = \sum_{\pi \in C} q_\lambda(\pi)[f(\pi) - f^*] = \frac{\sum_{\pi \in C} e^{-f(\pi)/\lambda}[f(\pi) - f^*]}{\sum_{\pi \in C} e^{-f(\pi)/\lambda}}. \quad (11)$$

The summations over $C$ in the numerator and the denominator can be split into two parts. Observing that $f(\pi) - f^* = 0$ for all $\pi \in \Pi^*$, we factor out $e^{-f^*/\lambda}$ from both the numerator and denominator in (11) and obtain that

$$\mathbb{E}_{q_\lambda}[f(\pi) - f^*] = \frac{\sum_{\pi \in \Pi^*} e^{-(f(\pi)-f^*)/\lambda}[f(\pi) - f^*] + \sum_{\pi \in C \backslash \Pi^*} e^{-(f(\pi)-f^*)/\lambda}[f(\pi) - f^*]}{\sum_{\pi \in \Pi^*} e^{-(f(\pi)-f^*)/\lambda} + \sum_{\pi \in C \backslash \Pi^*} e^{-(f(\pi)-f^*)/\lambda}}$$
$$= \frac{\sum_{\pi \in C \backslash \Pi^*} e^{-(f(\pi)-f^*)/\lambda}[f(\pi) - f^*]}{|\Pi^*| + \sum_{\pi \in C \backslash \Pi^*} e^{-(f(\pi)-f^*)/\lambda}}. \quad (12)$$

Reviewing that $\Delta := \min_{\pi \in C \backslash \Pi^*} f(\pi) - f^*$ represents the smallest gap between the objective values of suboptimal solutions and the optimal value, we have $f(\pi) - f^* \geq \Delta > 0$ for all $\pi \in C \backslash \Pi^*$. Given that the derivative of $xe^{-\lambda x}$ is $(1 - x/\lambda)e^{-x/\lambda}$, it follows that $xe^{-x/\lambda}$ is non-increasing for $x \geq \lambda$. Consequently, when $\Delta \geq \lambda > 0$, the condition $f(\pi) - f^* \geq \Delta \geq \lambda$ implies that $e^{-(f(\pi)-f^*)/\lambda}[f(\pi) - f^*] \leq \Delta e^{-\Delta/\lambda}$ for all $\pi \in C \backslash \Pi^*$. The numerator in (12) is bounded as

$$\sum_{\pi \in C \backslash \Pi^*} e^{-(f(\pi)-f^*)/\lambda}[f(\pi) - f^*] \leq |C \backslash \Pi^*|\Delta e^{-\Delta/\lambda} \leq |C|\Delta e^{-\Delta/\lambda}, \quad \text{for } \Delta \geq \lambda > 0. \quad (13)$$

Meanwhile, the denominator in (12) has the lower bound $|\Pi^*| + \sum_{\pi \in C \backslash \Pi^*} e^{-(f(\pi)-f^*)/\lambda} \geq |\Pi^*|$. It follows from (12) that

$$\mathbb{E}_{q_\lambda}[f(\pi) - f^*] \leq \frac{|C|\Delta e^{-\Delta/\lambda}}{|\Pi^*|}, \quad \text{for } \Delta \geq \lambda > 0. \quad (14)$$

4) Finally, combining these inequalities (8), (10) and (14), we derive that

$$\mathbb{P}_{p_{\theta^*(\lambda)}}(f(\pi) \geq f^* + \epsilon) \leq \frac{|C| \Delta e^{-\Delta/\lambda}}{|\Pi^*| \max\{\epsilon, \Delta\}} + \sqrt{\frac{c}{2\lambda}}, \quad \text{for } \Delta \geq \lambda > 0. \tag{15}$$

This completes the proof. □

## E EXPERIMENTS

### E.1 DATA GENERATION

#### E.1.1 TSPTW

The difficulty of a TSPTW instance largely depends on the availability of feasible solutions. Instances are more challenging when they allow very few feasible tours, making it harder for algorithms to find valid solutions, while overly feasible instances may lead the policy to overemphasize optimization and neglect feasibility. A key determinant of difficulty is the width of the time windows assigned to customer nodes. Narrower time windows reduce the overlap between customer service intervals, significantly limiting the number of feasible tours and increasing problem complexity.

To address these challenges, we adopt a data generation mechanism capable of producing TSPTW instances with easy, medium, and hard levels of difficulty. These levels are determined by systematically controlling the time window width and other instance-specific parameters, as detailed below:

**Easy and medium TSPTW instances.** An expected distance $T_n$ is predefined, which varies depending on the number of nodes, $n$. The ready times $e_i$ are sampled from a uniform distribution $\mathcal{U}[0, T_n]$. Time window widths $h_i$ are sampled from a scaled uniform distribution $\mathcal{U}[\alpha, \beta] \cdot T_n$, where $0 < \alpha < \beta < 1$ are hyperparameters. The due times $l_i$ are then calculated as $l_i = e_i + h_i$. By adjusting the hyperparameters $\alpha$, $\beta$, and $T_n$, TSPTW instances with varying levels of time window tightness can be generated. Specifically, to generate easy TSPTW instances, we set $\alpha = 0.5$ and $\beta = 0.75$. For medium TSPTW instances, we use $\alpha = 0.1$ and $\beta = 0.2$. In all cases, customer nodes are sampled uniformly within the region $[0, 100]^2$, and $T_n$ is set to $55(n + 1)$.

**Hard TSPTW instances.** Consistent with prior works (Kool et al., 2022; Bi et al., 2024), we adopt a generation mechanism inspired by benchmark datasets (Dumas et al., 1995; Da Silva & Urrutia, 2010) for hard TSPTW instances. This approach ensures the existence of at least one feasible tour. The core data generation procedure begins by sampling customer locations and constructing a random tour. Time windows are then defined centered around the arrival times within this random tour. Specifically, customer coordinates are uniformly sampled from $[0, 50]^2$, followed by the generation of a random permutation $\pi$ over customers (including the depot) to define a directed route. For each customer node $\pi_t$ in this sequence, the cumulative travel distance $d_{\pi_t}$ from the depot to $\pi_t$ along the random tour is calculated. Time windows are then constructed around $d_{\pi_t}$ with a pre-specified maximum width $w$. In benchmark datasets (Dumas et al., 1995; Da Silva & Urrutia, 2010), the maximum time window width $w$ typically ranges from 20 to 100. In this study, we select $w = 100$ . The ready time $e_{\pi_t}$ is sampled from $d_{\pi_t} - \mathcal{U}[0, w/2]$ and clamped to ensure non-negativity. The due time $l_{\pi_t}$ is similarly sampled from $d_{\pi_t} + \mathcal{U}[0, w/2]$.

For the depot, the ready time is conventionally set to zero, while the due time is left unconstrained since it does not affect the feasibility of the TSPTW solution, provided the depot remains reachable from all customer nodes.

**Normalization**. To standardize the input and ensure consistency across instances, node coordinates are scaled to the range $[0, 1]$ by dividing them by the maximum sample range. Time windows are proportionally scaled following this transformation to maintain consistency with the adjusted spatial coordinates.

#### E.1.2 TSPDL

Node coordinates are uniformly sampled from $[0, 1]^2$. Following (Rakke et al., 2012), we assign a demand of $q_0 = 0$ to the depot and a demand of $q_i = 1$ to each port node. Given a percentage $\sigma\%$, we randomly select $\lfloor (n + 1) \times \sigma\% \rfloor$ nodes with restricting draft limits strictly less than the total demand $\sum_{i=0}^{n} q_i$. Specifically, the draft limits of these selected nodes are randomly generated between 1 and

$n-1$. The remaining nodes are assigned a draft limit equal to the total demand $\sum_{i=0}^{n} q_i$, so that their draft limits impose no effective constraints.

To check whether the generated instance admits a feasible solution, we employ the following proposition from (Rakke et al., 2012) as a sufficient and necessary condition. If the condition is not satisfied, the instance is rejected and regenerated until a feasible one is obtained.

**Proposition E.1.** *(Rakke et al., 2012) Let $\pi = (\pi_1 = 0, \pi_2, \pi_3, \ldots, \pi_{n+1})$ be a solution ordering the port nodes $\{1, \ldots, n\}$ in ascending order of draft limits: $D_{\pi_2} \leq D_{\pi_3} \leq \ldots D_{\pi_{n+1}}$. Then the TSPDL instance admits a feasible solution if and only if $\pi$ is feasible.*

A larger percentage $\sigma\%$ results in more nodes with restricting draft limits, thereby increasing the difficulty of balancing feasibility and optimality. Following (Bi et al., 2024), we set $\sigma\%$ to $75\%$ for medium TSPDL datasets and $90\%$ for hard ones.

## E.2 IMPLEMENTATION DETAILS

**Baseline details.** For search-based solvers, including PyVRP, LKH3, VSR-LKH and OR-Tools, we run them with 32 CPU cores in parallel as done in (Kool et al., 2018; Zhou et al., 2024). For PIP, PIP-D and LMask, we run them on a single GPU with batch size of 2500 at inference time. Below we provide implementation details for each baseline.

- PyVRP, a state-of-the-art VRP solver built on top of HGS. We use the default hyperparameters and set a time limit of 20 seconds per instance for $n = 50$, and 50 seconds for $n = 100$. Note that PyVRP does not support draft limits and is therefore inapplicable to TSPDL.

- LKH3, a strong solver that implements the Lin-Kernighan heuristic for a wide range of routing problems. For each instance, we run LKH with 10000 trials and 1 run.

- OR-Tools, a more versatile solver than PyVRP and LKH3. For TSPTW, we use the local cheapest insertion as the first solution strategy and the guided local search as the local search strategy. As in PyVRP, we set the time limit to 20 seconds for $n = 50$, and 50 seconds for $n = 100$. Despite exhaustive testing across all initial solution strategies, OR-Tools fails to find any feasible solution for TSPDL within the time limit, consistent with findings in (Bi et al., 2024).

- PIP and PIP-D, state-of-the-art neural solvers for TSPTW and TSPDL. To maintain consistency with our proposed method and facilitate a fair comparison, we specifically utilize their implementations based on POMO. We evaluate PIP and PIP-D using the pretrained models and default hyperparameters as provided in their official source code repositories.

- VSR-LKH, a method combining reinforcement learning with LKH3 to solve TSP variants. We compile the official source code using its default settings, including the $\alpha$-measure for candidate selection.

- Random heuristics, two simple random heuristics with backtracking implemented to demonstrate the importance of the policy parameterized by a neural network. Random-L constructs a probability distribution by normalizing the inverse distances from the current node to candidate nodes. Random-C constructs a probability distribution by normalizing the inverse of due times in TSPTW or draft limits in TSPDL. The sample size is set to 64 and the batch size is accordingly set to 1024 for $n = 50$ and 512 for $n = 100$ due to memory limit.

**Overestimation initialization details.** As mentioned in the main text, we initialize the overestimation sets using problem-dependent lookahead strategies, namely single-step lookahead (SSL) and two-step lookahead (TSL). For TSPTW, according to the triangle inequality property of travel times, if an unvisited node violates the time window constraint at the current step, it will remain infeasible in subsequent steps without backtracking. Hence, SSL checks whether any unvisited node exhibits a time window violation. If such a node exists, $\hat{S}(\pi_{1:t})$ is initialized as empty, triggering backtracking; otherwise, it contains all unvisited nodes. TSL enhances this by tentatively extending the route with each candidate node and checking for time window violations among the remaining nodes. Nodes whose selection would make some remaining nodes infeasible are excluded, resulting in more accurate initialization and a notable decrease in backtracking steps. The TSPDL exhibits a similar monotonic property. Since the cumulative load increases along a route, any node violating its draft

limit at a given step will also be infeasible thereafter. Consequently, SSL and TSL for TSPDL are implemented analogously by checking for immediate and future draft limit violations, respectively.

**Hyperparameter details.** To ensure a fair comparison, we set the backtracking budget $R$ of LMask such that its runtime is comparable to that of neural baselines. The specific backtracking budgets used for experiments in Tables 1 and 2 are summarized in Table 4 . Our implementation is built upon the RL4CO library (Berto et al., 2023). Details of model and training hyperparameters of our main experiments are reported in Table 5. Note that the total number of gradient updates is comparable to that of neural baselines.

Table 4: Backtracking budget settings across different datasets.

| Hardness | TSPTW | | TSPDL | |
|---|---|---|---|---|
| | $n = 50$ | $n = 100$ | $n = 50$ | $n = 100$ |
| Easy | 100 | 150 | — | — |
| Medium | 100 | 200 | 150 | 150 |
| Hard | 200 | 300 | 150 | 150 |

Table 5: Experiment hyperparameters. Values with "/" indicate different choices depending on the problem size, i.e., on the left are values for $n = 50$ and on the right are values for $n = 100$.

| Hyperparameter | Value |
|---|---|
| *Model* | |
| Embedding dimension | 128 |
| Number of attention heads | 8 |
| Number of encoder layers | 6 |
| Normalization | Instance |
| Feedforward hidden dimension | 512 |
| Feedforward structure | MLP |
| Feedforward activation | ReLU |
| Tanh clipping | 10.0 |
| Refinement intensity feature dimension | 7 |
| *Training* | |
| Train decode type | Multi-sampling with free starts |
| Number of samples per instance | 50 / 100 |
| Batch size | 512 / 64 |
| Training instances per epoch | 256,000 / 100,000 |
| Penalty parameter $\rho$ | 1 |
| *Optimization* | |
| Optimizer | AdamW |
| Learning rate | 3e-4 / 1e-4 |
| Weight decay | 1e-6 |
| LR scheduler | MultiStepLR |
| LR milestones | [900, 950] |
| LR gamma | 0.1 |
| Gradient clip value | 1.0 |
| Max epochs | 1000 |

### E.3    Discussion on backtracking and overestimation set initialization

To thoroughly analyze the individual roles of backtracking and the overestimation set initialization strategy, as well as their interaction, we present results on medium TSPTW-50 for all 16 combinations of their usage at training and inference in Figure 5. Note that RIE is disabled in this experiment to ensure a fair comparison. We observe that, under TSL, enabling backtracking during training consistently reduces the optimality gap across inference settings. While training with backtracking can affect the model's inherent constraint awareness, this is effectively compensated by using backtracking during inference. As shown in Figure 5, when applying backtracking with the TSL strategy at inference, enabling backtracking during training leads to a notable reduction in the optimality gap (from 2.69% to 2.31%) with only a negligible increase in the solution infeasibility rate (from 0.05% to 0.14%). Moreover, this slight trade-off is mitigated by our proposed refinement intensity embedding, which further reduces the solution infeasibility rate to 0.05% and the optimality gap to 1.68%. We further observe that enabling backtracking at inference consistently yields substantial reductions in infeasibility and modest improvements in solution quality, regardless of the training configuration. Remarkably, using backtracking with less accurate SSL outperforms using TSL without backtracking, highlighting the critical role of inference-time backtracking.

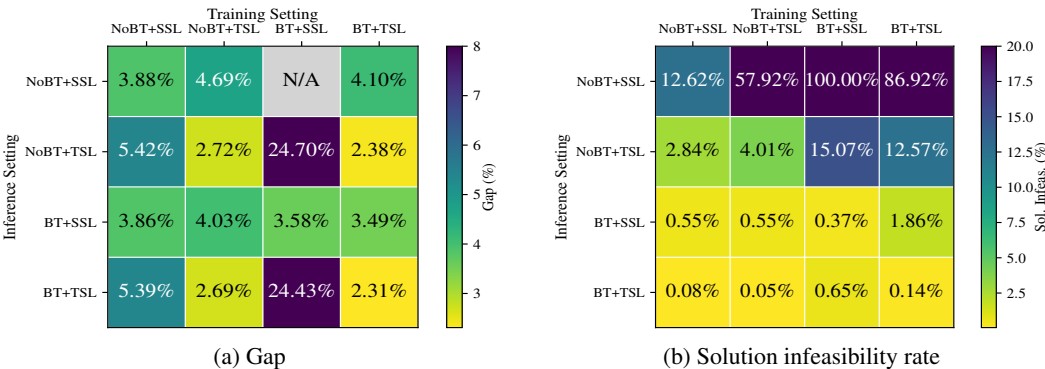

(a) Gap  (b) Solution infeasibility rate

Figure 5: Effect of backtracking and overestimation set initialization strategy combinations. BT/NoBT denotes with/without backtracking and SSL/TSL denotes single/two-step lookahead.

### E.4    Analysis of the Efficiency and Failure Modes of LMask

In this subsection, we provide an in-depth analysis of the circumstances under which LMask is efficient and those in which it is not. Recall that precisely obtaining the potential set $S$ is sometimes computationally intractable. Fortunately, our goal is solely to construct feasible solutions, and we use a pretrained policy to guide the node selection. Since this policy is trained with an $\ell_1$ penalty and its distribution is expected to approximate the feasible region well, we do not require the exact potential set $S$. Our approach is to establish an overestimation set $\hat{S}$ and refine it via backtracking when necessary. With the greedy decoding strategy, a feasible solution can be constructed as long as the node with the highest probability in the distribution $p_\theta(\cdot \mid \pi_{1:t})$ induced by $\hat{S}(\pi_{1:t})$, falls within the true potential set $S(\pi_{1:t})$. If at some step $t$, the policy incorrectly selects a node $i \in \hat{S}(\pi_{1:t}) \setminus S(\pi_{1:t})$, at a subsequent construction step, the algorithm will discover that the overestimation set for the corresponding extended route is empty, halting further progress. At this point, our backtracking mechanism is triggered. Through this repeated process of extension and backtracking, the algorithm eventually corrects the incorrect selection at step $t$ and removes node $i$ from $\hat{S}(\pi_{1:t})$.

Based on this analysis, two key factors influence the efficiency of the LMask algorithm can be identified. The first is the correspondence between the high-probability nodes in the distribution $p_\theta(\cdot \mid \pi_{1:t})$ induced by $\hat{S}(\pi_{1:t})$ and the true potential set $S(\pi_{1:t})$ at each decoding step $t$. The second is the cost of correcting an incorrect selection $i \in \hat{S}(\pi_{1:t}) \setminus S(\pi_{1:t})$ at decoding step $t$. This cost is related to both the magnitude of $t$ and the initial overestimation set $\hat{S}([\pi_{1:t}, i])$ corresponding to the extended route $[\pi_{1:t}, i]$. The former determines the backtracking depth, while the latter determines the number of exploration attempts.

To illustrate these factors, we visualize the LazyMask decoding process on two representative small-scale TSPDL instances ($n = 19$). In the first example (Figure 6), LMask efficiently constructs a feasible solution with a relatively small backtracking budget. Although the gap between the true potential sets and the overestimation sets is large in the early decoding steps, it has no impact because the policy accurately identifies nodes within the potential set. In a later decoding step ($t = 14$), the policy incorrectly selects a node outside the true potential set. However, because this error occurs late, the backtracking depth is not large, and the number of nodes in the potential set of the extended route is small. Therefore, a backtracking budget of only $R = 100$ is sufficient to correct this error and ultimately construct a feasible solution. In the second example (Figure 7), LMask requires a huge backtracking budget to successfully construct a feasible solution. Here, the initial overestimation set at each step is very close to the true potential set, differing by only one node. However, the policy selects this single incorrect node at an early step ($t = 9$), and the overestimation set for the corresponding extended route is large. Consequently, a significant backtracking budget is spent attempting to correct this early error (even $R = 10$k was insufficient). By increasing the budget to $R = 11$k, LMask is able to correct this selection and finally constructs a feasible solution.

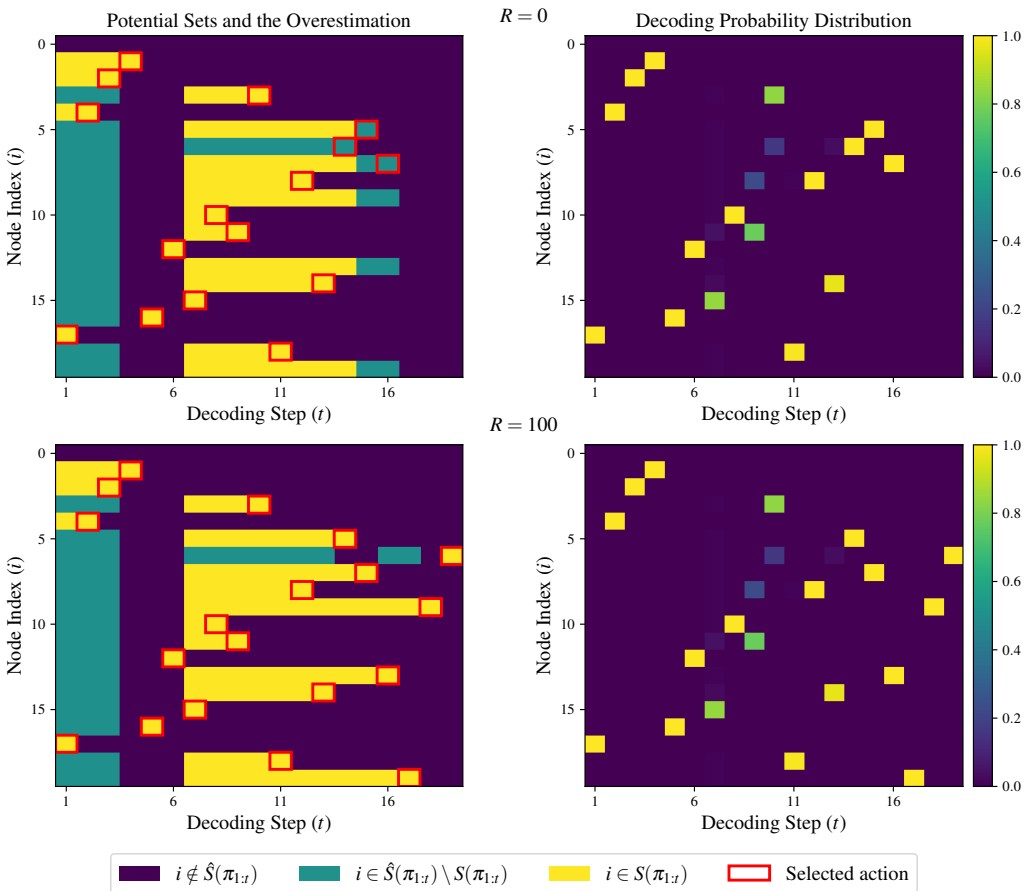

Figure 6: Visualization of the LazyMask decoding process on a TSPDL instance where a feasible solution is efficiently found with a small backtracking budget. The rows depict two decoding trials with varying backtracking budgets: $R = 0$ (Top) and $R = 100$ (Bottom). The trial with $R = 0$ fails to find a feasible solution, whereas the trial with $R = 100$ successfully identifies a complete, feasible route. **Left:** The left column visualizes the relationship between the true potential set $S(\pi_{1:t})$ and its overestimation $\hat{S}(\pi_{1:t})$. At each decoding step $t$ (x-axis), each node $i$ (y-axis) is color-coded based on its set membership. Dark blue indicates $i \notin \hat{S}(\pi_{1:t})$, teal green indicates $i \in \hat{S}(\pi_{1:t}) \setminus S(\pi_{1:t})$, and bright yellow indicates $i \in S(\pi_{1:t})$. The node selected by the policy at each step is indicated by a red bounding box. The visualization for any failed trial is truncated at the decoding step where $S(\pi_{1:t})$ becomes empty and the backtracking budget is depleted. **Right:** The right column displays the corresponding decoding probability distribution $p_\theta(\cdot \mid \pi_{1:t})$ generated by the LMask at each step.

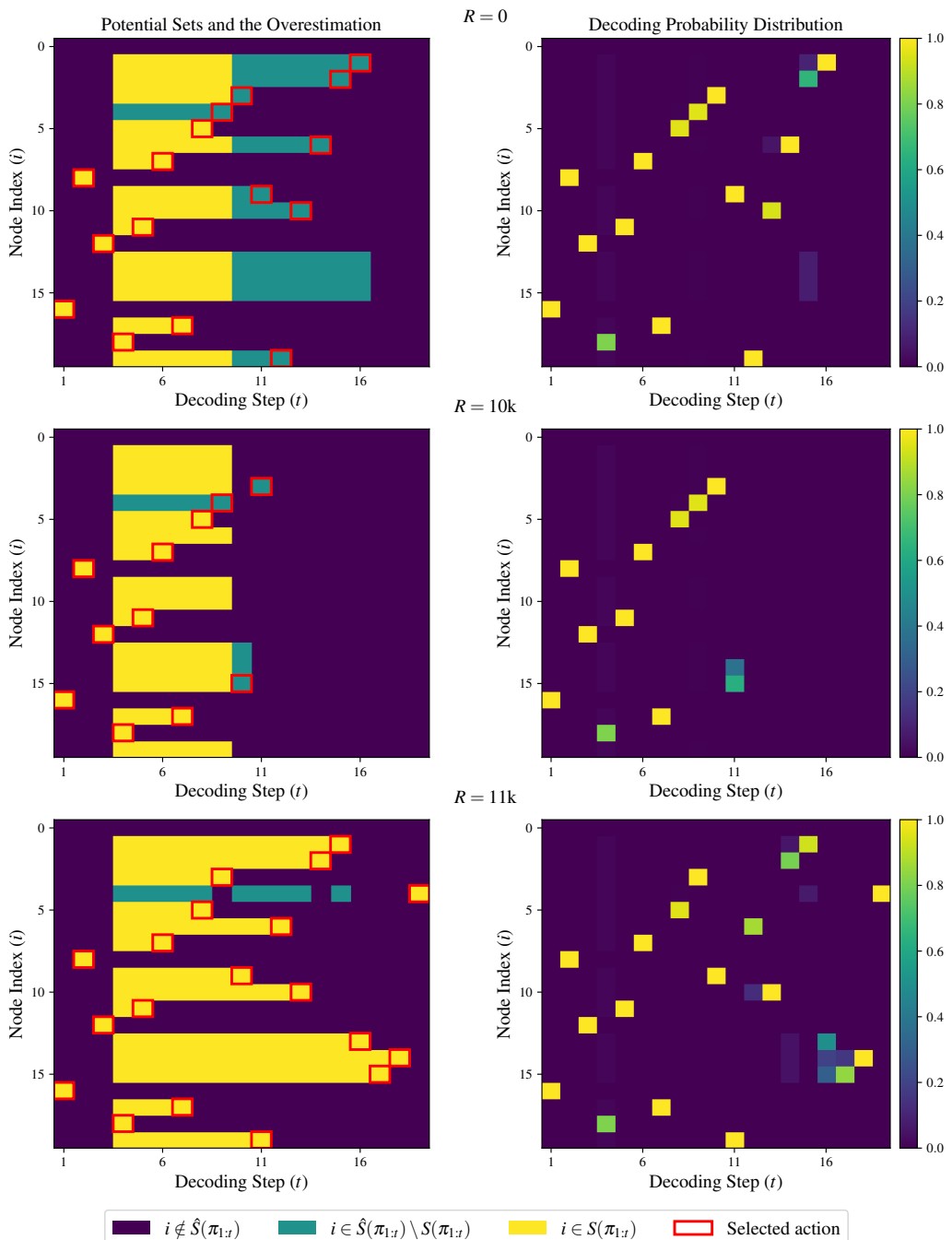

Figure 7: Visualization of the LazyMask decoding process on a challenging TSPDL instance requiring extensive backtracking. The rows depict three decoding trials with varying backtracking budgets: $R = 0$ (Top), $R = 10k$ (Middle), and $R = 11k$ (Bottom). The first two trials ($R = 0, R = 10k$) fail to find a feasible solution, whereas the $R = 11k$ trial successfully identifies a complete, feasible route. **Left:** The left column visualizes the relationship between the true potential set $S(\pi_{1:t})$ and its overestimation $\hat{S}(\pi_{1:t})$. At each decoding step $t$ (x-axis), each node $i$ (y-axis) is color-coded based on its set membership. Dark blue indicates $i \notin \hat{S}(\pi_{1:t})$, teal green indicates $i \in \hat{S}(\pi_{1:t}) \setminus S(\pi_{1:t})$, and bright yellow indicates $i \in S(\pi_{1:t})$. The node selected by the policy at each step is indicated by a red bounding box. The visualization for any failed trial is truncated at the decoding step where $S(\pi_{1:t})$ becomes empty and the backtracking budget is depleted. **Right:** The right column displays the corresponding decoding probability distribution $p_\theta(\cdot \mid \pi_{1:t})$ generated by the LMask at each step.

## E.5 Effect of the penalty parameter.

We analyze the sensitivity of LMask's performance to the $\ell_1$ penalty parameter $\rho$. The results on the hard TSPDL-50 dataset are presented in Table 6. In our experiments, we tested several fixed values for $\rho$ and a scheduled approach, denoted as $\rho^\dagger$. For the scheduled approach, the penalty parameter is increased exponentially over the training epochs $t$ according to the schedule $\rho_t = \min(\gamma^t \cdot \rho_0, \rho_{max})$, with the hyperparameters set to $\gamma = 1.01$, $\rho_0 = 0.5$, and $\rho_{max} = 2$. Our results show that a small, fixed penalty ($\rho = 0.5$) leads to training instability, resulting in high infeasibility and a large optimality gap. Conversely, large fixed penalty values (e.g., $\rho = 1.5$, $\rho = 2$) reduce infeasibility but at the cost of an increased optimality gap. The scheduled approach ($\rho^\dagger$) provides better training stability and strikes an effective balance, achieving low infeasibility rates and a small optimality gap that slightly outperform our default setting of $\rho = 1$.

Table 6: Results on the hard TSPDL-50 dataset with different $\rho$ during training.

| $\rho$ | Sol. Infeas. | Ins. Infeas. | Obj. | Gap |
|---|---|---|---|---|
| 0.5 | 7.10% | 2.45% | 13.71 | 3.97% |
| 1 | 0.19% | 0.04% | 13.57 | 2.52% |
| 1.5 | 0.07% | 0.03% | 13.58 | 2.64% |
| 2 | 0.02% | 0.01% | 13.60 | 2.78% |
| $\rho^\dagger$ | **0.18%** | **0.00%** | **13.56** | **2.48%** |

## E.6 Generalization and scalability

### E.6.1 Generalization across time window widths

We assess the generalization ability of LMask on hard TSPTW-50 datasets, by evaluating performance on test instances with varying maximum time window widths $w = 20, 60, 200$, while keeping the training distribution fixed at $w = 100$. As shown in Table 7, both neural baselines, PIP and PIP-D, exhibit large performance fluctuations as $w$ deviates from $100$. In particular, although their performance improves at $w = 20$, their infeasibility rates and optimality gaps increase significantly at $w = 60$ and $w = 200$, compared to their performance on $w = 100$ reported in Table 1 of the main paper. This indicates limited generalization when the maximum time width changes. In contrast, LMask remains consistently robust across different values of $w$, showing zero infeasibility rates in all cases, even outperforming the strong traditional solver PyVRP in feasibility. While its optimality gap increases at $w = 200$, it remains lower than those of PIP and PIP-D. These results demonstrate the strong generalization ability of LMask to variations in the maximum time window width.

### E.6.2 Scalability to large-scale problem instances

To assess the scalability of LMask, we conduct experiments on the hard TSPTW-500 dataset, which contains 128 hard TSPTW instances with $n = 500$. We do not include PIP and PIP-D in this evaluation as their official source code repository does not offer pretrained models specifically for the hard TSPTW-500 dataset. Due to the substantial computational cost per instance at this scale, traditional solvers such as PyVRP, LKH3, and OR-Tools are no longer able to process multiple instances in parallel using multi-core CPUs as done in the main experiments. Instead, each instance is solved sequentially using all available 32 CPU cores. Therefore, we report the average runtime per instance for a fair comparison. As for the size-specific hyperparameters, we set the time limit of PyVRP and OR-Tools for solving each instance to 4 minutes. LKH3 is run with a maximum of 10000 trials and 10 runs. LMask's inference is performed with a backtracking budget of $R = 600$.

As shown in Table 8, OR-Tools and both greedy heuristics fail to produce any feasible solutions at this scale, with instance infeasibility rates reaching $100\%$. In contrast, traditional solvers such as PyVRP and LKH still manage to find feasible solutions for the majority of instances, achieving low

infeasibility rates of $3.12\%$ and $2.33\%$, respectively. Notably, LMask achieves zero infeasibility rates and maintains a low optimality gap of $0.53\%$, demonstrating superior scalability.

Table 7: Results on datasets with varying maximum time window widths $w$. All models are trained on hard TSPTW instances with $n = 50$ and $w = 100$.

| Width | $w = 20$ | | | $w = 60$ | | | $w = 200$ | | |
|---|---|---|---|---|---|---|---|---|---|
| Method | Infeasible | | Gap | Infeasible | | Gap | Infeasible | | Gap |
| | Sol. | Ins. | | Sol. | Ins. | | Sol. | Ins. | |
| PyVRP | - | 12.6% | * | - | 2.71% | * | - | 0.03% | * |
| PIP | 3.75% | 2.54% | 0.08% | 6.54% | 3.64% | 0.04% | 10.49% | 5.69% | 3.57% |
| PIP-D | 3.08% | 1.82% | 0.08% | 8.72% | 4.67% | 0.08% | 18.15% | 10.35% | 6.58% |
| LMask | **0.00%** | **0.00%** | **0.00%** | **0.00%** | **0.00%** | **0.01%** | **0.00%** | **0.00%** | **2.99%** |

Table 8: Results on hard TSPTW-500

| Method | Sol. Infeas. | Ins. Infeas. | Obj. | Gap | Avg. Time |
|---|---|---|---|---|---|
| PyVRP | - | 3.12% | 256.59 | * | 4m |
| LKH | - | 2.33% | 256.65 | 0.00% | 13m |
| OR-Tools | - | 100.00% | - | - | 17s |
| Greedy-L | 100.00% | 100.00% | - | - | 1s |
| Greedy-C | 100.00% | 100.00% | - | - | 1s |
| LMask | 0.00% | 0.00% | 257.92 | 0.53% | 16s |

### E.7 PERFORMANCE OF LMASK USING DIFFERENT BACKBONE MODELS

LMask is model-agnostic and can be combined with any auto-regressive backbone model. The default backbone model used in main experiments is POMO (Kwon et al., 2020). Here we replace it with the recent ReLD model (Huang et al., 2025), which enhances the POMO decoder by adding residual connections and a two-layer feed-forward network. The performance comparison of different backbone models on various datasets is presented in Table 9. The results show that a stronger backbone model further boosts the performance of LMask.

Table 9: Performance comparison of LMask using different backbone models.

| Nodes | | | $n = 50$ | | | | | $n = 100$ | | | | |
|---|---|---|---|---|---|---|---|---|---|---|---|---|
| Dataset | Hardness | Model | Infeasible | | Obj. | Gap | Time | Infeasible | | Obj. | Gap | Time |
| | | | Sol. | Inst. | | | | Sol. | Inst. | | | |
| TSPTW | Easy | POMO | 0.06% | **0.00%** | 7.45 | 2.02% | 7s | **0.01%** | **0.00%** | 10.50 | 3.11% | 17s |
| | | ReLD | **0.01%** | **0.00%** | **7.43** | **1.66%** | 7s | 0.02% | **0.00%** | **10.48** | **2.85%** | 21s |
| | Medium | POMO | 0.04% | **0.00%** | 13.25 | 1.68% | 6s | 0.05% | **0.00%** | 19.51 | 4.23% | 18s |
| | | ReLD | **0.03%** | **0.00%** | **13.23** | **1.58%** | 7s | **0.03%** | **0.00%** | **19.31** | **3.13%** | 22s |
| | Hard | POMO | **0.00%** | **0.00%** | 25.71 | 0.10% | 6s | **0.00%** | **0.00%** | 51.38 | 0.21% | 18s |
| | | ReLD | **0.00%** | **0.00%** | **25.70** | **0.06%** | 7s | **0.00%** | **0.00%** | **51.36** | **0.16%** | 22s |
| TSPDL | Medium | POMO | **0.03%** | 0.01% | 11.14 | 2.75% | 6s | 0.20% | 0.05% | 17.04 | 4.24% | 15s |
| | | ReLD | **0.03%** | **0.00%** | **11.06** | **2.00%** | 7s | **0.18%** | **0.03%** | **16.91** | **3.42%** | 15s |
| | Hard | POMO | 0.19% | 0.04% | 13.57 | 2.52% | 6s | 0.80% | 0.26% | 21.63 | 4.34% | 15s |
| | | ReLD | **0.13%** | **0.02%** | **13.47** | **1.80%** | 7s | **0.52%** | **0.11%** | **21.53** | **3.84%** | 16s |

### E.8 SAMPLING PERFORMANCE

We evaluate the performance of neural methods under sampling decoding. Each method samples $S$ solutions per augmentation, where $S$ is varied as 3, 10, and 30. The $8\times$ dihedral augmentation is

retained, resulting in $8 \times S$ solutions per instance. Due to memory limitations, the batch sizes are adjusted accordingly. For an intuitive comparison, we also include the results under greedy decoding as reported in the main paper. The results are summarized in Tables 10 and 11.

Compared to greedy decoding, all methods exhibit lower instance infeasibility rates and optimality gaps under sampling decoding. Meanwhile, the solution infeasibility rates increase under sampling decoding, as the larger number of sampled solutions naturally leads to a higher proportion of infeasible ones. Remarkably, even though PIP and PIP-D generate ten times as many solutions as the greedy version of LMask, they still yield higher instance infeasibility rates. In addition, their optimality gaps are generally higher than those of greedy LMask on most datasets, highlighting LMask's strong ability to consistently generate high-quality feasible solutions even with a smaller number of samples.

Table 10: Sampling results on synthetic TSPTW datasets.

| Nodes | | $n = 50$ | | | | | $n = 100$ | | | | |
|---|---|---|---|---|---|---|---|---|---|---|---|
| **Method** | **Infeasible** | | **Obj.** | **Gap** | **Time** | **Infeasible** | | **Obj.** | **Gap** | **Time** |
| | **Sol.** | **Inst.** | | | | **Sol.** | **Inst.** | | | |
| | PIP | 0.28% | 0.01% | 7.51 | 2.73% | 9s | 0.16% | 0.00% | 10.57 | 3.78% | 29s |
| | PIP ($S = 3$) | 0.29% | 0.01% | 7.49 | 2.45% | 19s | 0.17% | 0.00% | 10.55 | 3.56% | 1.3m |
| | PIP ($S = 10$) | 0.29% | 0.01% | 7.47 | 2.20% | 57s | 0.17% | 0.00% | 10.52 | 3.24% | 4.6m |
| | PIP ($S = 30$) | 0.29% | 0.01% | 7.45 | 2.03% | 3.0m | 0.18% | 0.00% | 10.50 | 3.04% | 12.8m |
| Easy | PIP-D | 0.28% | 0.00% | 7.50 | 2.60% | 10s | 0.05% | 0.00% | 10.66 | 4.62% | 31s |
| | PIP-D ($S = 3$) | 0.31% | 0.00% | 7.48 | 2.33% | 20s | 0.07% | 0.00% | 10.64 | 4.40% | 1.5m |
| | PIP-D ($S = 10$) | 0.29% | 0.00% | 7.46 | 2.06% | 1.1m | 0.06% | 0.00% | 10.60 | 4.05% | 5.0m |
| | PIP-D ($S = 30$) | 0.30% | 0.00% | 7.44 | 1.89% | 3.3m | 0.06% | 0.00% | 10.58 | 3.82% | 14.0m |
| | LMask | 0.06% | 0.00% | 7.45 | 2.02% | 7s | 0.01% | 0.00% | 10.50 | 3.11% | 17s |
| | LMask ($S = 3$) | 0.06% | 0.00% | 7.44 | 1.85% | 11s | 0.02% | 0.00% | 10.53 | 3.31% | 35s |
| | LMask ($S = 10$) | 0.07% | 0.00% | 7.42 | 1.51% | 17s | 0.02% | 0.00% | 10.48 | 2.89% | 1.2m |
| | LMask ($S = 30$) | 0.06% | 0.00% | 7.40 | 1.30% | 33s | 0.02% | 0.00% | 10.45 | 2.61% | 3.1m |
| | PIP | 4.82% | 1.07% | 13.41 | 2.93% | 10s | 4.35% | 0.39% | 19.61 | 4.79% | 29s |
| | PIP ($S = 3$) | 4.98% | 0.68% | 13.36 | 2.55% | 19s | 4.55% | 0.19% | 19.54 | 4.37% | 1.3m |
| | PIP ($S = 10$) | 4.96% | 0.46% | 13.32 | 2.26% | 58s | 4.52% | 0.10% | 19.45 | 3.93% | 4.5m |
| | PIP ($S = 30$) | 4.98% | 0.25% | 13.30 | 2.06% | 2.9m | 4.50% | 0.06% | 19.39 | 3.60% | 12.7m |
| Medium | PIP-D | 4.14% | 0.90% | 13.46 | 3.31% | 9s | 3.46% | 0.03% | 19.80 | 5.76% | 31s |
| | PIP-D ($S = 3$) | 4.30% | 0.54% | 13.41 | 2.95% | 20s | 3.87% | 0.00% | 19.73 | 5.41% | 1.5m |
| | PIP-D ($S = 10$) | 4.32% | 0.36% | 13.37 | 2.66% | 1.1m | 3.80% | 0.00% | 19.63 | 4.86% | 5.0m |
| | PIP-D ($S = 30$) | 4.31% | 0.27% | 13.34 | 2.44% | 3.3m | 3.82% | 0.00% | 19.55 | 4.47% | 14.0m |
| | LMask | 0.06% | 0.00% | 13.25 | 1.73% | 6s | 0.05% | 0.00% | 19.51 | 4.23% | 18s |
| | LMask ($S = 3$) | 0.08% | 0.00% | 13.30 | 2.13% | 14s | 0.10% | 0.00% | 19.48 | 4.05% | 40s |
| | LMask ($S = 10$) | 0.07% | 0.00% | 13.23 | 1.60% | 24s | 0.10% | 0.00% | 19.40 | 3.64% | 1.3m |
| | LMask ($S = 30$) | 0.07% | 0.00% | 13.19 | 1.25% | 55s | 0.11% | 0.00% | 19.34 | 3.33% | 3.2m |
| | PIP | 5.65% | 2.85% | 25.73 | 0.18% | 9s | 31.74% | 16.68% | 51.48 | 0.37% | 28s |
| | PIP ($S = 3$) | 5.81% | 2.28% | 25.72 | 0.16% | 19s | 32.47% | 11.83% | 51.44 | 0.33% | 1.3m |
| | PIP ($S = 10$) | 5.80% | 1.89% | 25.72 | 0.15% | 58s | 32.59% | 9.66% | 51.43 | 0.30% | 4.5m |
| | PIP ($S = 30$) | 5.79% | 1.68% | 25.72 | 0.14% | 2.9m | 32.58% | 7.97% | 51.42 | 0.27% | 12.7m |
| Hard | PIP-D | 6.44% | 3.03% | 25.75 | 0.27% | 9s | 13.59% | 6.60% | 51.43 | 0.32% | 31s |
| | PIP-D ($S = 3$) | 6.59% | 2.40% | 25.75 | 0.25% | 20s | 13.99% | 5.32% | 51.41 | 0.29% | 1.5m |
| | PIP-D ($S = 10$) | 6.56% | 2.10% | 25.74 | 0.24% | 1.1m | 13.93% | 4.54% | 51.41 | 0.27% | 5.0m |
| | PIP-D ($S = 30$) | 6.58% | 1.88% | 25.74 | 0.23% | 3.3m | 13.92% | 4.05% | 51.40 | 0.26% | 14.0m |
| | LMask | 0.00% | 0.00% | 25.71 | 0.10% | 6s | 0.00% | 0.00% | 51.38 | 0.21% | 18s |
| | LMask ($S = 3$) | 0.00% | 0.00% | 25.70 | 0.09% | 10s | 0.00% | 0.00% | 51.37 | 0.19% | 40s |
| | LMask ($S = 10$) | 0.00% | 0.00% | 25.70 | 0.08% | 17s | 0.00% | 0.00% | 51.36 | 0.18% | 1.3m |
| | LMask ($S = 30$) | 0.00% | 0.00% | 25.70 | 0.08% | 33s | 0.00% | 0.00% | 51.36 | 0.16% | 3.2m |

### E.9 STABILITY ANALYSIS

We investigate the stability of LMask under sampling decoding at inference. In Figure 8, we present box plots of the optimality gaps and solution infeasibility rates across 10 different random seeds on medium TSPTW-50 and TSPDL-50 datasets. Across both datasets and all tested random seeds, both metrics remain highly stable, with total variations staying below $0.02\%$. These results demonstrate

Table 11: Sampling results on synthetic TSPDL datasets.

| Nodes | | $n = 50$ | | | | | $n = 100$ | | | | |
| --- | --- | --- | --- | --- | --- | --- | --- | --- | --- | --- | --- |
| **Method** | | **Infeasible** | | **Obj.** | **Gap** | **Time** | **Infeasible** | | **Obj.** | **Gap** | **Time** |
| | | Sol. | Inst. | | | | Sol. | Inst. | | | |
| **Medium** | PIP | 1.75% | 0.17% | 11.23 | 3.59% | 8s | 2.50% | 0.16% | 17.68 | 8.10% | 21s |
| | PIP ($S = 3$) | 1.87% | 0.13% | 11.19 | 3.23% | 15s | 2.68% | 0.10% | 17.62 | 7.68% | 52s |
| | PIP ($S = 10$) | 1.84% | 0.11% | 11.16 | 2.93% | 45s | 2.68% | 0.08% | 17.53 | 7.14% | 2.9m |
| | PIP ($S = 30$) | 1.84% | 0.11% | 11.14 | 2.73% | 2.5m | 2.70% | 0.06% | 17.46 | 6.74% | 8.8m |
| | PIP-D | 2.29% | 0.22% | 11.26 | 3.96% | 8s | 1.83% | 0.23% | 17.80 | 8.84% | 23s |
| | PIP-D ($S = 3$) | 2.42% | 0.10% | 11.22 | 3.56% | 17s | 1.98% | 0.14% | 17.73 | 8.40% | 59s |
| | PIP-D ($S = 10$) | 2.44% | 0.07% | 11.19 | 3.19% | 50s | 1.95% | 0.09% | 17.63 | 7.78% | 3.4m |
| | PIP-D ($S = 30$) | 2.45% | 0.07% | 11.16 | 2.94% | 2.8m | 1.96% | 0.07% | 17.56 | 7.36% | 10.0m |
| | LMask | 0.03% | 0.01% | 11.14 | 2.75% | 6s | 0.20% | 0.05% | 17.04 | 4.24% | 15s |
| | LMask ($S = 3$) | 0.04% | 0.01% | 11.10 | 2.43% | 13s | 0.21% | 0.04% | 16.99 | 3.88% | 28s |
| | LMask ($S = 10$) | 0.04% | 0.01% | 11.07 | 2.15% | 17s | 0.21% | 0.04% | 16.91 | 3.43% | 51s |
| | LMask ($S = 30$) | 0.04% | 0.01% | 11.05 | 1.97% | 29s | 0.21% | 0.04% | 16.86 | 3.13% | 2.0m |
| **Hard** | PIP | 4.83% | 2.39% | 13.63 | 3.42% | 8s | 29.34% | 21.65% | 22.35 | 12.87% | 20s |
| | PIP ($S = 3$) | 5.06% | 2.14% | 13.59 | 3.09% | 15s | 29.60% | 20.30% | 22.30 | 12.37% | 52s |
| | PIP ($S = 10$) | 5.07% | 2.00% | 13.56 | 2.83% | 44s | 29.52% | 19.53% | 22.18 | 11.59% | 3.0m |
| | PIP ($S = 30$) | 5.07% | 1.90% | 13.54 | 2.64% | 2.5m | 29.52% | 18.92% | 22.08 | 11.00% | 8.8m |
| | PIP-D | 4.16% | 0.82% | 13.79 | 4.28% | 8s | 13.51% | 8.43% | 22.90 | 12.53% | 23s |
| | PIP-D ($S = 3$) | 4.43% | 0.59% | 13.74 | 3.89% | 17s | 13.76% | 7.49% | 22.82 | 11.99% | 1.0m |
| | PIP-D ($S = 10$) | 4.47% | 0.40% | 13.70 | 3.55% | 51s | 13.77% | 6.82% | 22.71 | 11.30% | 3.4m |
| | PIP-D ($S = 30$) | 4.43% | 0.38% | 13.67 | 3.32% | 2.8m | 13.75% | 6.55% | 22.63 | 10.80% | 10.0m |
| | LMask | 0.19% | 0.04% | 13.57 | 2.52% | 6s | 0.80% | 0.26% | 21.63 | 4.34% | 15s |
| | LMask ($S = 3$) | 0.22% | 0.04% | 13.53 | 2.22% | 13s | 0.87% | 0.22% | 21.56 | 3.99% | 29s |
| | LMask ($S = 10$) | 0.22% | 0.03% | 13.50 | 1.98% | 18s | 0.87% | 0.21% | 21.48 | 3.61% | 52s |
| | LMask ($S = 30$) | 0.22% | 0.03% | 13.48 | 1.82% | 29s | 0.87% | 0.18% | 21.42 | 3.32% | 2.0m |

that LMask yields highly consistent performance when using sampling decoding at inference, despite the inherent randomness in solution generation.

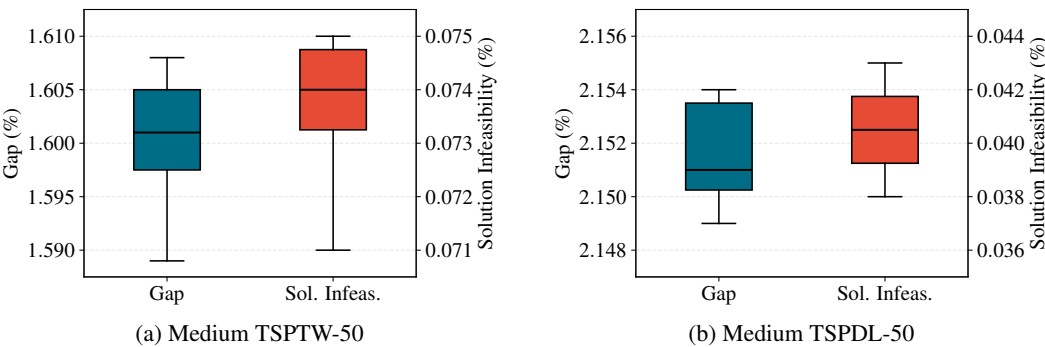

(a) Medium TSPTW-50

(b) Medium TSPDL-50

Figure 8: Box plots of optimality gaps and solution infeasibility rates of LMask under sampling decoding across 10 random seeds. Each box shows the interquartile range (25th-75th percentile), the horizontal line indicates the median, and the whiskers extend to the minimum and maximum within 1.5 times the interquartile range.

### E.10 PERFORMANCE ON THE TSPTW BENCHMARK

We evaluate our LMask framework on the TSPTW benchmark (Dumas et al., 1995) to validate our innovation. Although comprehensive tests are conducted on all TSPTW benchmark instances, we decide to directly reference the original test results from Appendix D.7 in the article (Bi et al., 2024) due to inaccurate replication attempts of PIP and PIP-D. Additionally, to ensure consistency in research comparisons, the benchmark instance set we selected remains fully aligned with those

used in the original study. Results show that compared to the neural baselines, LMask framework significantly reduces the infeasibility rate and improves solution quality.

Table 12: Model performance on the benchmark datasets (Dumas et al., 1995).

| Instance | Opt. | PIP | | PIP-D | | LMask | | Instance | Opt. | PIP | | PIP-D | | LMask | |
|---|---|---|---|---|---|---|---|---|---|---|---|---|---|---|---|
| | | Obj. | Gap | Obj. | Gap | Obj. | Gap | | | Obj. | Gap | Obj. | Gap | Obj. | Gap |
| n20w20.001 | 378 | 389 | 2.91% | 389 | 2.91% | 378 | 0.00% | n40w40.003 | 474 | 496 | 4.64% | 497 | 4.85% | 474 | 0.00% |
| n20w20.002 | 286 | 292 | 2.10% | 292 | 2.10% | 286 | 0.00% | n40w40.004 | 452 | - | - | - | - | 460 | 1.77% |
| n20w20.003 | 394 | - | - | - | - | 394 | 0.00% | n40w40.005 | 453 | 470 | 3.75% | 471 | 3.97% | 458 | 1.10% |
| n20w20.004 | 396 | 405 | 2.27% | 405 | 2.27% | 396 | 0.00% | n40w60.001 | 494 | - | - | 525 | 6.28% | 504 | 2.02% |
| n20w20.005 | 352 | 360 | 2.27% | 360 | 2.27% | 352 | 0.00% | n40w60.002 | 470 | - | - | 502 | 6.81% | 493 | 4.89% |
| n20w40.001 | 254 | 276 | 8.66% | 279 | 9.84% | 257 | 1.18% | n40w60.003 | 408 | - | - | - | - | 414 | 1.47% |
| n20w40.002 | 333 | 347 | 4.20% | 339 | 1.80% | 333 | 0.00% | n40w60.004 | 382 | 406 | 6.28% | 420 | 9.95% | 393 | 2.88% |
| n20w40.003 | 317 | 332 | 4.73% | 332 | 4.73% | 317 | 0.00% | n40w60.005 | 328 | 342 | 4.27% | 344 | 4.88% | 332 | 1.22% |
| n20w40.004 | 388 | 401 | 3.35% | 401 | 3.35% | 389 | 0.26% | n40w80.001 | 395 | 407 | 3.04% | 407 | 3.04% | 395 | 0.00% |
| n20w40.005 | 288 | 294 | 2.08% | 302 | 4.86% | 289 | 0.35% | n40w80.002 | 431 | 448 | 3.94% | 452 | 4.87% | 433 | 0.46% |
| n20w60.001 | 335 | 349 | 4.18% | 353 | 5.37% | 336 | 0.30% | n40w80.003 | 412 | 444 | 7.77% | 454 | 10.19% | 416 | 0.97% |
| n20w60.002 | 244 | 252 | 3.28% | 260 | 6.56% | 246 | 0.82% | n40w80.004 | 417 | 430 | 3.12% | 435 | 4.32% | 421 | 0.96% |
| n20w60.003 | 352 | 358 | 1.70% | 358 | 1.70% | 352 | 0.00% | n40w80.005 | 344 | 362 | 5.23% | 379 | 10.17% | 344 | 0.00% |
| n20w60.004 | 280 | 298 | 6.43% | 289 | 3.21% | 282 | 0.71% | n60w80.001 | 458 | - | - | - | - | 461 | 0.66% |
| n20w60.005 | 338 | 385 | 13.91% | 361 | 6.80% | 340 | 0.59% | n60w80.002 | 498 | 540 | 8.43% | 548 | 10.04% | 522 | 4.82% |
| n20w80.001 | 329 | 347 | 5.47% | 347 | 5.47% | 330 | 0.30% | n60w80.003 | 550 | 635 | 15.45% | 646 | 17.45% | 592 | 7.64% |
| n20w80.002 | 353 | 347 | 2.66% | 360 | 6.51% | 338 | 0.00% | n60w80.004 | 566 | 611 | 7.95% | 632 | 11.66% | 587 | 3.71% |
| n20w80.003 | 320 | 328 | 2.50% | 328 | 2.50% | 320 | 0.00% | n60w80.005 | 468 | 535 | 14.32% | - | - | 491 | 4.91% |
| n20w80.004 | 304 | 341 | 12.17% | 339 | 11.51% | 324 | 6.58% | n80w60.001 | 554 | 582 | 5.05% | - | - | 563 | 1.62% |
| n20w80.005 | 264 | 302 | 14.39% | 302 | 14.39% | 272 | 3.03% | n80w60.002 | 633 | 678 | 7.11% | - | - | 647 | 2.21% |
| n40w20.001 | 500 | - | - | - | - | 500 | 0.00% | n80w60.004 | 619 | 678 | 9.53% | - | - | 642 | 3.72% |
| n40w20.002 | 552 | - | - | 610 | 10.51% | 552 | 0.00% | n80w60.005 | 575 | - | - | - | - | 600 | 4.35% |
| n40w20.003 | 478 | - | - | 507 | 6.07% | 478 | 0.00% | n80w80.001 | 624 | - | - | - | - | 643 | 3.04% |
| n40w20.004 | 404 | 419 | 3.71% | 418 | 3.47% | 407 | 0.74% | n80w80.002 | 592 | 624 | 5.41% | 638 | 7.77% | 614 | 3.72% |
| n40w20.005 | 499 | - | - | - | - | 501 | 0.40% | n80w80.003 | 589 | 648 | 10.02% | 674 | 14.43% | 623 | 5.77% |
| n40w40.001 | 465 | - | - | - | - | 465 | 0.00% | n80w80.004 | 594 | 674 | 13.47% | 676 | 13.80% | 619 | 4.21% |
| n40w40.002 | 461 | 485 | 5.21% | 483 | 4.77% | 470 | 1.95% | n80w80.005 | 570 | 627 | 10.00% | - | - | 585 | 2.63% |
| **Average Gap** | | 5.2% | | 5.3% | | **0.6%** | | **Average Gap** | | 7.4% | | 8.5% | | **2.6%** | |
| **Infeasible Rate** | | 22.2% | | 14.8% | | **0.0%** | | **Infeasible%** | | 25.9% | | 37.9% | | **0.0%** | |

## F    RUNTIME DISCUSSION

### F.1    INFERENCE-TIME EVALUATION PROTOCOL AND CPU-ONLY RESULTS

It is hard to achieve an absolutely fair comparison of the run time between CPU-based traditional solvers and GPU-based solvers due to hardware difference. As a widely adopted convention in the neural combinatorial optimization community, we have run traditional solvers with 32 CPUs in parallel to mitigate this unfairness. To provide further clarification, we have also run LMask using only CPUs. The results, presented in Table 13, demonstrate that LMask remains substantially faster than the traditional solvers.

### F.2    BACKTRACKING OVERHEAD

The runtime of LMask can be decomposed into four components: (1) network forward pass; (2) lookahead; (3) backtracking; and (4) miscellaneous operations such as distance matrix precomputation and tour length evaluation. We then elaborate on the overhead induced by backtracking. The main source of cost comes from reduced parallelization due to instance heterogeneity and from increased forward passes in the decoder. The backtracking operation itself is lightweight, as it only rolls back via a stack. To mitigate the cost from heterogeneity, one can selectively process only the instances still in forward construction. In fact, this approach is used during the inference phase with greedy decoding. As shown in Figure 2(b), on the hard TSPTW-100 dataset, increasing the backtracking budget from 0 to 100 adds only about 2 seconds to the inference time, which is a marginal increase. However, during the training phase, which employs a multi-sampling decoding scheme, this selective processing would impair the computational efficiency of the cross-attention mechanism in the decoder. We believe that more efficient implementations could further reduce this training overhead.

To clearly demonstrate the computational overhead from backtracking and compare training times with other methods, we provide a training time comparison in Table R5. Note that we have optimized the implementation of lookahead from the original PIP source code by reducing the use of indexing operations, which are very time-consuming on GPUs. This optimization significantly improves

efficiency. As a result, even with the addition of backtracking, the total training time for LMask remains shorter than that of PIP.

Table 13: Inference time using 32 CPUs. Time is reported in minutes (m), hours (h), and days (d).

| Hardness | Method | $n = 50$ | $n = 100$ |
|----------|--------|----------|-----------|
| Easy | PyVRP | 1.7h | 4.3h |
| | LKH3 | 1.9h | 7.2h |
| | LMask | 3.2m | 11.4m |
| Medium | PyVRP | 1.7h | 4.3h |
| | LKH3 | 2.9h | 10.3h |
| | LMask | 3.8m | 11.7m |
| Hard | PyVRP | 1.7h | 4.3h |
| | LKH3 | 2.3h | 1.3d |
| | LMask | 5.6m | 12.4m |

Table 14: Training time (days).

| Method | $n = 50$ | $n = 100$ |
|--------|----------|-----------|
| LMask | 4.2d | 7.8d |
| PIP | 4.7d | 28.4d |
| PIP-D | 3.6d | 12.2d |

## G  LICENSES FOR EXISTING ASSETS

The used assets in this work are listed in Table 15, which are all open-source for academic research. We will release our source code with the MIT License.

Table 15: Used assets, licenses, and their usage.

| Type | Asset | License | Usage |
|------|-------|---------|-------|
| Code | LKH3 (Helsgaun, 2017) | Available for academic use | Evaluation |
| | OR-Tools (Furnon & Perron, 2024) | Apache-2.0 license | Evaluation |
| | PyVRP (Furnon & Perron, 2024) | MIT License | Evaluation |
| | RL4CO (Furnon & Perron, 2024) | MIT License | Revision |
| | PIP (Furnon & Perron, 2024) | MIT License | Revision |
| Datasets | Dumas et al. (Dumas et al., 1995) | Available for academic use | Evaluation |

## H  THE USE OF LARGE LANGUAGE MODELS

Large language models are used only for writing.

