# OpenReview forum: "LMask: Learn to Solve Constrained Routing Problems with Lazy Masking"
_ICLR.cc/2026/Conference — ICLR 2026 Poster_

### Official Review · Reviewer_KHo7 · 2025-10-25

**Soundness:** 3
**Presentation:** 3
**Contribution:** 2
**Rating:** 4
**Confidence:** 4

**Summary:**

This paper presents LazyMask (LMask), a new decoding mechanism coupled with refinement intensity embedding (RIE) that augments model state with search-trace information for solving hard-constrained routing problems, specifically TSPTW and TSPDL. The key design is integrating a backtracking-based masking strategy that enables neural constructive methods to generate feasible solutions more effectively. Results on TSPTW and TSPDL show that LMask achieves lower infeasibility rates and improved solution quality compared to baselines.

**Strengths:**

- The paper is generally well-written with clear presentation and logical structure.
- The proposed LMask represents a meaningful advancement for neural constructive methods applied to TSPTW and TSPDL.
- The inclusion of theoretical analysis provides valuable insights.
- The experimental evaluation is extensive and the results are promising.

**Weaknesses:**

- The applicability of the Lmask seems limited. The discussions and experiments focus almost entirely on TSPTW and TSPDL, both of which involve temporal constraints. It is uncertain whether the backtracking-based masking strategy and RIE mechanism can be effectively extended to other complex constrained VRPs with different constraints.
- The paper lacks an in-depth analysis of the gap between the true potential set $S$ and its estimation $\hat{S}$. There is no empirical or theoretical investigation of how well such approximations work under different conditions.
- The selection of the backtracking budget $R$ appears empirical without principled guidelines.
- Infeasibility rates in Tables 1 and 2 are still not 0.00%, while classical methods like LKH3 achieve perfect feasibility. The paper should investigate why the Lmask still fails in those cases and what additional methods may be needed to fully close this gap.

**Questions:**

Please see my comments above.

---

> ### Author Response · Authors · 2025-11-22
>
> Thanks for your careful and valuable comments. Our changes and responses to the review’s questions are listed below. We sincerely hope that the clarifications and revisions will address all of your concerns and allow you to reconsider the rating.
>
> > **W1. The applicability of the Lmask seems limited. The discussions and experiments focus almost entirely on TSPTW and TSPDL, both of which involve temporal constraints. It is uncertain whether the backtracking-based masking strategy and RIE mechanism can be effectively extended to other complex constrained VRPs with different constraints.**
>
> We follow the established convention in the literature [1,2] by focusing on TSP variants. These variants serve as standard benchmarks for learning-based methods tackling combinatorial optimization problems with complex hard constraints. This choice ensures comparability and clearly isolates our methodological contributions. Furthermore, the TSP variants we consider are inherently challenging due to their complex constraints. For instance, even finding a feasible solution for the TSPTW is known to be NP-hard [3]. Although TSPTW can be viewed as a special case of the vehicle routing problem with time windows (VRPTW), state-of-the-art algorithms for VRPTW are often not well suited for TSPTW and may experience extreme degradation on this specific problem [4].
>
> Nevertheless, our LMask framework is applicable in principle to a broader class of routing problems with complex constraints, extending beyond time windows and draft limits. A primary example is the vehicle routing problem with time windows and flexible delivery locations (VRPTW-FL). In this extension of VRPTW, each customer can be served at one of several potential locations rather than a single fixed one. Each location has a capacity limit on the number of customers served at the same time. These capacity limitations impose global constraints across different routes, making it challenging to construct a feasible solution using a one-pass forward framework. In contrast, the backtracking mechanism used in LMask has been proven efficient in [5] for revising early decisions that lead to future infeasibility in this problem. We believe that within the learning framework of LMask, a pre-trained policy guiding node selection will further enhance the efficiency of this backtracking mechanism.
>
>
> [1] Chen, Jingxiao, et al. "Looking ahead to avoid being late: Solving hard-constrained traveling salesman problem." arXiv preprint arXiv:2403.05318 (2024).
>
> [2] Jieyi Bi, Yining Ma, Jianan Zhou, Wen Song, Zhiguang Cao, Yaoxin Wu, and Jie Zhang. "Learning to handle complex constraints for vehicle routing problems." In Advances in Neural Information Processing Systems, 2024.
>
> [3] Martin WP Savelsbergh. "Local search in routing problems with time windows." Annals of Operations research, 4:285–305, 1985.
>
> [4] Dumas, Yvan, et al. "An optimal algorithm for the traveling salesman problem with time windows." Operations research 43.2 (1995): 367-371.
>
> [5] Frey, Christian MM, et al. "The vehicle routing problem with time windows and flexible delivery locations." European Journal of Operational Research 308.3 (2023): 1142-1159.

---

> ### Author Response · Authors · 2025-11-22
>
> > **W2. The paper lacks an in-depth analysis of the gap between the true potential set $S$ and its estimation $\hat{S}$. There is no empirical or theoretical investigation of how well such approximations work under different conditions.**
>
> As stated in our paper, precisely obtaining the true potential set $S$ for routing problems with complex constraints is sometimes computationally intractable. However, our goal is solely to construct feasible solutions, and we use a pre-trained policy to guide the node selection during the construction process. Since this policy is trained with an $\ell_1$ penalty and its distribution is expected to approximate the feasible region well, we do not strict require the exact potential set $S$.
> Our approach is to establish an overestimation set $\hat{S}$ of $S$ and refine this set via the backtracking mechanism when necessary. Taking the greedy decoding strategy as an example, ideally, a feasible solution can be constructed as long as the node with the highest probability in  the distribution $p_{\theta}(\cdot\mid\pi_{1:t})$ induced by $\hat{S}(\pi_{1:t})$, falls within the true potential set $S(\pi_{1:t})$. If at some step $t$, the policy incorrectly selects a node $i \in \hat{S}(\pi_{1:t})\setminus S(\pi_{1:t})$, at a subsequent construction step, the algorithm will discover that the overestimation set for the corresponding extended path is empty, halting further progress. At this point, our backtracking mechanism is triggered. Through this repeated process of extension and backtracking, the algorithm eventually corrects the incorrect selection at step $t$ and removes node $i$ from the overestimation set $\hat{S}(\pi_{1:t})$.
>
> Based on this analysis, several key factors influence the efficiency of the LMask algorithm can be identified:
> - At each decoding step $t$, the correspondence between the high-probability nodes in the distribution $p_{\theta}(\cdot \mid \pi_{1:t})$ induced by $\hat{S}(\pi_{1:t})$ and the true potential set $S(\pi_{1:t})$.
> - The cost of correcting an incorrect selection $i\in \hat{S}(\pi_{1:t})\setminus S(\pi_{1:t})$ at decoding step $t$. This cost is related to both the magnitude of $t$ and the overestimation set $\hat{S}([\pi_{1:t},i])$ corresponding to the extended path $[\pi_{1:t}, i]$. The former determines the backtracking depth, while the latter determines the number of exploration attempts.
>
> To deepen understanding, we have added a visualization of the LazyMask decoding process on two small-scale instances in the newly uploaded paper (see Appendix E.5). The results are summarized as follows:
> - In the first example, LMask efficiently constructs a feasible solution with a relatively small backtracking budget. Although the gap between the true potential sets and the overestimation sets is large in the early decoding steps, it has no impact because the policy accurately identifies nodes within the potential set. In a later decoding step, the policy incorrectly selects a node outside the true potential set. However, because this error occurs late, the backtracking depth is not large, and the number of nodes in the potential set of the extended path is small. Therefore, a backtracking budget of only $R=100$ is sufficient to correct this error and ultimately construct a feasible solution.
> - In the second example, LMask requires a huge backtracking budget to successfully construct a feasible solution. In this instance, the initial overestimation set at each step is very close to the true potential set, differing by only one node. However, the policy selects this single incorrect node at an early decoding step, and the overestimation set for the corresponding extended path is large. Consequently, a significant backtracking budget is spent attempting to correct this early error (even $R=10$k was insufficient). By increasing the budget to $R=11$k, LMask is able to correct this selection and finally constructs a feasible solution.
>
> > **W3. The selection of the backtracking budget appears empirical without principled guidelines.**
>
> From the perspective of effectiveness, given the inherent difficulty of finding feasible solutions for routing problems with complex constraints, we acknowledge that a practical guideline for setting $R$ to guarantee feasibility is not available. However, from the perspective of computation time, given a time budget and the problem scale, we can estimate a suitable $R$ through preliminary experiments. For example, the results in Figure 4 of our paper show that on the hard TSPTW-100 dataset, the inference time increases approximately linearly with $R$. For every increase of 100 in $R$, the inference time increases by approximately 2-3 seconds.

---

> ### Author Response · Authors · 2025-11-22
>
> > **W4. Infeasibility rates in Tables 1 and 2 are still not 0.00%, while classical methods like LKH3 achieve perfect feasibility. The paper should investigate why the Lmask still fails in those cases and what additional methods may be needed to fully close this gap.**
>
> LMask is a constructive method with neural network, whereas LKH3 is an iterative search method. Note that LKH3 searches a far greater number of solutions than the 8 solutions constructed by LMask, and it may also search infeasible solutions. Therefore, a fairer comparison is the instance infeasibility rate. On all TSPTW datasets in Table 1, LMask achieves zero instance infeasibility rates. Notably, on hard TSPTW-100, LMask demonstrates superior instance feasibility compared to the strong traditional solver LKH3, which does not achieve perfect instance feasibility on this dataset.
>
> However, there are indeed some instances that require extensive backtracking for LMask to construct a feasible solution. To balance feasibility and computational efficiency, we set the backtracking budget to a level where LMask's inference time is comparable to that of the neural baselines. This is why LMask did not achieve perfect instance feasibility rates in Table 2. The reason for LMask's failure on these instances has been provided in our analysis for W2. To close this gap, one could consider increasing the backtracking budget or employing additional post-processing methods, such as local search, to repair feasibility.

---

### Official Review · Reviewer_V3Tu · 2025-10-31

**Soundness:** 3
**Presentation:** 3
**Contribution:** 3
**Rating:** 6
**Confidence:** 4

**Summary:**

This paper proposes LMask, a novel method for enhancing auto-regressive neural solvers on constrained routing problems. Specifically, it integrates a backtracking mechanism during decoding to prevent from infeasible solutions, and designs a refinement intensity embedding method to mitigate representation ambiguity induced by backtracking. The validity of the proposed method is theoretically analyzed under specific assumptions. Experiments on two typical constrained routing problems, TSPTW and TSPDL, demonstrate its superior performance over existing methods.

**Strengths:**

1. The proposed method is well-motivated. The paper is well-written and easy to follow.
2. On the two benchmark problems TSPTW and TSPDL, the results clearly demonstrate the superiority of LMask against competitors.
3. The authors provide extensive ablation and hyperparameter studies, with clear analysis.

**Weaknesses:**

1. The meaning of the theoretical parts is limited, since the conditions and assumptions seems much too strict.

**Questions:**

1. How does LMask perform when paired with other architectures, such as LEHD?

---

> ### Author Response · Authors · 2025-11-22
>
> Thanks for your careful and valuable comments. Our changes and responses to the review’s questions are listed below. We sincerely hope that the clarifications and revisions will address all of your concerns.
>
> > **W1. The meaning of the theoretical parts is limited, since the conditions and assumptions seems much too strict.**
>
> 1. Proposition 4.1 relies on the assumption of an infinite computational budget $R=+\infty$. This assumption is reasonable because for constrained routing problems where finding a feasible solution is known to be NP-complete (e.g., TSPTW), guaranteeing feasibility within polynomial time is theoretically impossible unless P=NP. Proposition 4.1 aims to establish the asymptotic completeness of the proposed decoding method, proving that the algorithm possesses the intrinsic capability to find a feasible solution. This analysis provides a theoretical foundation regarding the algorithm's limit performance and soundness, distinguishing LMask from prior constructive neural solvers that lack such guarantees, even though practical implementations naturally impose a finite budget for efficiency.
> 2. Theorem 4.3 relies on the assumption of the neural network's approximation capability, which is a challenging topic in deep learning theory. However, this assumption serves merely to quantify the representational capacity of the neural network. We argue that this condition is not prohibitively strict and does not diminish the practical value of Theorem 4.3, which extends beyond the bound itself. Its primary contribution lies in theoretically characterizing the intrinsic trade-off governed by the entropy regularization coefficient $\lambda$, which provides concrete guidance for hyperparameter tuning. Theorem 4.3 explicitly decomposes the performance bound into two competing terms, revealing a fundamental conflict between concentration and approximation:
> -   Small $\lambda$ (Strong Concentration): A smaller $\lambda$ enhances the term $e^{-\Delta(\mathcal{P})/\lambda}$, leading to stronger exponential suppression of suboptimal solutions. However, this comes at the cost of increasing the approximation error term $\sqrt{\frac{c}{2\lambda}}$. This aligns with the intuition that a sharper distribution is theoretically optimal but significantly harder for a neural network to approximate accurately.
> -   Large $\lambda$ (Easier Approximation): Conversely, a larger $\lambda$ reduces the approximation error term, implying that a smoother (high-entropy) distribution is easier to learn. However, this weakens the probability guarantee, leading to a more dispersed search space.
>
> > **Q1: How does LMask perform when paired with other architectures, such as LEHD?**
>
> The default backbone model in our experiments is POMO. In Table 10 of the Appendix, we have presented results from replacing POMO with the recent ReLD model. ReLD enhances the POMO decoder by adding residual connections and a two-layer feed-forward network. This idea is similar to that of LEHD, as both aim to enhance the decoder's capability. However, ReLD is more lightweight and, on the problem scales we considered, more effective. To intuitively demonstrate the impact of the backbone model on LMask, we re-present these results in **Table R3**. The results show that a stronger backbone model further boosts the performance of LMask.
>
> **Table R3: Performance comparison of LMask using different backbone models.**
> | | | | | | **$n=50$** | | | | | **$n=100$** | | |
> | :--- | :--- | :--- | :--- | :--- | :--- | :--- | :--- | :--- | :--- | :--- | :--- | :--- |
> | **Dataset** | **Hardness** | **Model** | **Sol. Infeas.** | **Inst. Infeas.** | **Obj.** | **Gap** | **Time** | **Sol. Infeas.** | **Inst. Infeas.** | **Obj.** | **Gap** | **Time** |
> | **TSPTW** | **Easy** | POMO | 0.06% | **0.00%** | 7.45 | 2.02% | 7s | **0.01%** | **0.00%** | 10.50 | 3.11% | 17s |
> | | | ReLD | **0.01%** | **0.00%** | **7.43** | **1.66%** | 7s | 0.02% | **0.00%** | **10.48** | **2.85%** | 21s |
> | | **Medium** | POMO | 0.04% | **0.00%** | 13.25 | 1.68% | 6s | 0.05% | **0.00%** | 19.51 | 4.23% | 18s |
> | | | ReLD | **0.03%** | **0.00%** | **13.23** | **1.58%** | 7s | **0.03%** | **0.00%** | **19.31** | **3.13%** | 22s |
> | | **Hard** | POMO | **0.00%** | **0.00%** | 25.71 | 0.10% | 6s | **0.00%** | **0.00%** | 51.38 | 0.21% | 18s |
> | | | ReLD | **0.00%** | **0.00%** | **25.70** | **0.06%** | 7s | **0.00%** | **0.00%** | **51.36** | **0.16%** | 22s |
> | **TSPDL** | **Medium** | POMO | 0.03% | 0.01% | 11.14 | 2.75% | 6s | 0.20% | 0.05% | 17.04 | 4.24% | 15s |
> | | | ReLD | **0.03%** | **0.00%** | **11.06** | **2.00%** | 7s | **0.18%** | **0.03%** | **16.91** | **3.42%** | 15s |
> | | **Hard** | POMO | 0.19% | 0.04% | 13.57 | 2.52% | 6s | 0.80% | 0.26% | 21.63 | 4.34% | 15s |
> | | | ReLD | **0.13%** | **0.02%** | **13.47** | **1.80%** | 7s | **0.52%** | **0.11%** | **21.53** | **3.84%** | 16s |

---

### Official Review · Reviewer_jBMF · 2025-10-31

**Soundness:** 3
**Presentation:** 3
**Contribution:** 3
**Rating:** 8
**Confidence:** 3

**Summary:**

The paper proposes LMask, a learning framework for constrained routing that decodes with a LazyMask mechanism combining lightweight lookahead with adaptive backtracking. It introduces refinement intensity embedding (RIE) to encode the backtracking trace into the decoder. Theoretical guarantees are also provided: with unbounded backtracking, LazyMask generates only feasible solutions and assigns non-zero probability to all feasible ones, and a probabilistic optimality bound is analyzed.

**Strengths:**

1.	The proposed LMask contributes significantly to the current masking mechanism, which excludes the infeasible actions perfectly and efficiently via a smart backtracking design.

2.	Proposition 4.1 provides a strong feasibility guarantee, lending theoretical grounding to masking-based constraint handling.

3.	The method achieves notable improvements on the TSPTW and TSPDL benchmarks, while maintaining competitive inference times and demonstrating robustness across instance scales and distributions

**Weaknesses:**

The theoretical contribution of Theorem 4.3 appears limited for practice. Its bound relies on an approximation assumption and does not directly inform the backtracking design or concrete hyperparameter choices, thus offering limited guidance for algorithmic tuning.

**Questions:**

Could the proposed backtracking mechanism be extended to large language model (LLM)-based solvers? While LLMs do not share the same MDP formalism, they are also autoregressive. I am curious about whether LazyMask-style backtracking, and RIE-like signals might transfer to general sequential models. This is only for discussion; no additional experiments are requested.

---

> ### Author Response · Authors · 2025-11-22
>
> Thanks for your careful and valuable comments. We sincerely hope that the clarifications will address all of your concerns.
>
> > **W1. The theoretical contribution of Theorem 4.3 appears limited for practice. Its bound relies on an approximation assumption and does not directly inform the backtracking design or concrete hyperparameter choices, thus offering limited guidance for algorithmic tuning.**
>
> We thank the reviewer for this keen observation. We acknowledge that our theoretical bound relies on the assumption of the neural network's approximation capability—a challenging aspect common to deep learning theory. However, it is more interesting that the practical value of Theorem 4.3 extends beyond the bound itself. Its primary contribution lies in theoretically characterizing the intrinsic trade-off governed by the entropy regularization coefficient $\lambda$, which provides concrete guidance for hyperparameter tuning. Theorem 4.3 explicitly decomposes the performance bound into two competing terms, revealing a fundamental conflict between concentration and approximation:
> -   Small $\lambda$ (Strong Concentration): A smaller $\lambda$ enhances the term $e^{-\Delta(\mathcal{P})/\lambda}$, leading to stronger exponential suppression of suboptimal solutions. However, this comes at the cost of increasing the approximation error term $\sqrt{\frac{c}{2\lambda}}$. This aligns with the intuition that a sharper distribution is theoretically optimal but significantly harder for a neural network to approximate accurately.
> -   Large $\lambda$ (Easier Approximation): Conversely, a larger $\lambda$ reduces the approximation error term, implying that a smoother (high-entropy) distribution is easier to learn. However, this weakens the probability guarantee, leading to a more dispersed search space.
>
>
> > **Q1. Could the proposed backtracking mechanism be extended to large language model (LLM)-based solvers? While LLMs do not share the same MDP formalism, they are also autoregressive. I am curious about whether LazyMask-style backtracking, and RIE-like signals might transfer to general sequential models. This is only for discussion; no additional experiments are requested.**
>
> We appreciate this insightful question. Extending the LazyMask-style backtracking and RIE-like singals to large language model (LLM)-based solvers is indeed a promising direction, as both are fundamentally auto-regressive.  To adapt our framework, the definition of the potential set $S$ must first be tailored to the specific problem scenario. For instance, while routing problems seek a feasible path under hard constraints, code generation tasks focus on syntax correctness, and general text generation may prioritize safety or factual consistency. Moreover, designing an efficient strategy to obtain the overestimation of $S$ is also crucial, as this directly determines the efficiency of backtracking in a large vocabulary space.
>
> Regarding your specific interest in LLM-based routing solvers, our mechanism fits naturally into this emerging domain. Since LLMs often struggle with strict constraints like time windows, LazyMask could serve as an inference-time intervention layer. When the model generates a token that violates constraints, meaning the potential set $S$ becomes empty, the system would trigger a backtrack rather than continuing an invalid path. Furthermore, the concept of RIE could be adapted into "soft prompts" or special embedding tokens. These signals would inform the LLM of the current search state, such as the number of retries, helping the model adjust its probability distribution to avoid repeating mistakes.
>
> The philosophy of backtracking is also emerging in general sequential modeling. For example, previous work [1] introduces a special `[RESET]` token, allowing the language model to realize mistakes and retract its generation. This implies that sequence models can indeed be trained to utilize backtracking behaviors effectively.
>
> [1] Yiming Zhang, Jianfeng Chi, Hailey Nguyen, Kartikeya Upasani, Daniel M. Bikel, Jason E Weston, and Eric Michael Smith. "Backtracking improves generation safety." ICLR, 2025.

---

> > ### Comment · Reviewer_jBMF · 2025-11-27
> >
> > Thank you for your insightful response. I will maintain my positive score.

---

### Official Review · Reviewer_nv24 · 2025-11-01

**Soundness:** 3
**Presentation:** 3
**Contribution:** 3
**Rating:** 6
**Confidence:** 5

**Summary:**

The paper aims to address the challenge of constrained routing problems. It introduces the LazyMask decoding method, which lazily refines feasibility masks with the backtracking mechanism. In addition, it employs the refinement intensity embedding to encode the search trace into the model. The proposed method is tested on TSPTW and TSPDL.

**Strengths:**

- This paper aims to address the challenging constrained routing problems in neural methods.
- The idea of using lazy masks to correct the decoding errors on the nonlinear routing problems is interesting.
- The performance on the complex TSPTW and TSPDL is good, with low infeasibility and competitive solution quality. The experiments are thorough.

**Weaknesses:**

- The backtracking mechanism introduces additional computational overhead.
- The idea is not that impressive. PIP adopts a lookahead mask, whereas LMask uses backtracking. Also, it would be better to theoretically and empirically analyse the computational complexity of these two methods. Why is LMask, with a two-step lookahead and backtracking, reported to be faster than PIP, which also uses a two-step lookahead but without backtracking?
- The selection criteria for the backtracking budget on each problem are unclear.
- The theoretical guarantee provided for LMask relies on the assumption of an infinite computational budget, which is impractical in real-world scenarios.
- The proposed method appears to be specifically tailored for constrained nonlinear problems. Can it also be applied to constrained linear problems?

**Questions:**

-  Is TSL initialization the same as PIP?
- What are Greedy-C and Greedy-C in Table 9?

**Details Of Ethics Concerns:**

I have no ethical concerns.

---

> ### Author Response · Authors · 2025-11-22
>
> Thanks for your careful and valuable comments. Our changes and responses to the review’s questions are listed below. We sincerely hope that the clarifications and revisions will address all of your concerns.
>
> > **W1. The backtracking mechanism introduces additional computational overhead.**
>
> While the backtracking mechanism inevitably introduces additional computational overhead, we argue that for NP-hard problems, an affordable increase in computational cost is justified to facilitate the discovery of feasible solutions. Compared to a backtracking-free one forward-pass decoding strategy, this additional computation significantly enhances the capability of the LMask framework to generate feasible solutions without incurring a prohibitive runtime cost. The computational overhead introduced by the backtracking mechanism is very small compared to its benefits. For example, on the hard TSPTW-100 dataset, LMask with the TSL overestimation initialization strategy and a backtracking budget of $R=300$ reduces the solution infeasibility rate from 15% to zero, while the inference time increases by only about 2 seconds. To help understand the empirical overhead from backtracking, we additionally provide the inference time of LMask at $R=0$ on various datasets in Table **R1**. These results clarify that backtracking adds only a marginal increase in runtime.
>
> **Table R1: Time comparison (inference time in seconds).**
> | | |  | $n=50$| |  | $n=100$| |
> | :--- | :--- | :--- | :--- | :--- | :--- | :--- | :--- |
> |**Problem** | **Method** | **Easy** | **Medium** | **Hard** | **Easy** | **Medium** | **Hard** |
> |  | PIP| 9 | 9 | 9 | 29 | 29 | 28 |
> | **TSPTW** | LMask ($R=0$) | 5 | 5 | 5 | 16 | 16 | 16 |
> | | LMask | 7 | 6 | 6 | 17 | 18 | 18 |
> |  | PIP| | 8 | 8 | | 20 | 20 |
> | **TSPDL** | LMask ($R=0$) | | 4 | 4 | | 12 | 12 |
> | | LMask | | 6 | 6 | | 15 | 15 |
>
>
> > **W2. The idea is not that impressive. PIP adopts a lookahead mask, whereas LMask uses backtracking. Also, it would be better to theoretically and empirically analyse the computational complexity of these two methods. Why is LMask, with a two-step lookahead and backtracking, reported to be faster than PIP, which also uses a two-step lookahead but without backtracking?**
>
> Theoretically, a 2-step lookahead typically incurs $O(n^2)$ complexity per step, whereas unconstrained backtracking can exhibit exponential complexity in the worst case. However, to ensure efficiency in our framework, we impose a strict budget, that is, a maximum of $R$ backtracking steps, to prevent exponential explosions. This effectively bounds the worst-case complexity of backtracking to be linear with respect to the budget $R$. In practice, the neural network is trained to generate high-quality initial proposals.
>
> To isolate the specific cost of the backtracking mechanism, we have conducted an ablation study comparing LMask with and without the backtracking one (see **Table R1** above). The reported PIP results are based on the source code from their official repository. In our LMask implementation, we optimized the efficiency of the lookahead by reducing the use of indexing operations, as clarified in Appendix F. The specific implementation difference is detailed in our answer to Q1. Moreover, the additional computational overhead introduced by backtracking is small. This combination results in LMask reporting faster times than PIP.
>
> > **W3. The selection criteria for the backtracking budget on each problem are unclear.**
>
> The backtracking budget serves as a hyperparameter strictly controlling the maximum computational overhead allowed during the decoding phase. Its value was determined empirically based on performance evaluations on the validation set. We selected the budget to achieve an optimal trade-off between computational efficiency and the feasibility rate. Regarding the backtracking setting, we empirically set the budget $R$ to a level that results in a final inference time comparable to that of the neural baselines. The specific backtracking budget for each problem has been presented in Table 5 in the Appendix. For clarity, we restate the specific backtracking budget settings here in **Table R2**.
>
> **Table R2: Backtracking budget settings across different datasets.**
> | |  | $n=50$| |  | $n=100$| |
> | :--- | :---: | :---: | :---: | :---: | :---: | :---: |
> | **Problem** | **Easy** | **Medium** | **Hard** | **Easy** | **Medium** | **Hard** |
> | **TSPTW** | 100 | 100 | 200 | 150 | 200 | 300 |
> | **TSPDL** | --- | 150 | 150 | --- | 150 | 150 |

---

> ### Author Response · Authors · 2025-11-22
>
> > **W4. The theoretical guarantee provided for LMask relies on the assumption of an infinite computational budget, which is impractical in real-world scenarios.**
>
> For constrained routing problems where finding a feasible solution is known to be NP-hard, such as TSPTW [1], guaranteeing feasibility within polynomial time is theoretically impossible unless P=NP. Proposition 4.1 aims to establish the asymptotic completeness of the proposed decoding method, proving that the algorithm possesses the intrinsic capability to find a feasible solution. This analysis provides a theoretical foundation regarding the algorithm's limit performance and soundness, distinguishing LMask from prior constructive neural solvers that lack such guarantees, even though practical implementations naturally impose a finite budget for efficiency.
>
> [1] Martin WP Savelsbergh. "Local search in routing problems with time windows." Annals of Operations research, 4:285–305, 1985.
>
>
> > **W5. The proposed method appears to be specifically tailored for constrained nonlinear problems. Can it also be applied to constrained linear problems?**
>
> The proposed method is not limited to the linearity or nonlinearity of the problem. Whether a problem is represented as linear or nonlinear depends on the mathematical modeling approach we choose. If we use sequential modeling for the routing problem, the objective is nonlinear with node variables in $\{0,1,\dots,n\}$. However, the mixed integer linear programming formulation employs the binary variables $x_{ij}$ to indicate whether a path is taken from node $i$ to node $j$. In this classic MILP formulation, both the objective function and the constraints are constructed as linear expressions of these binary variables. Furthermore, the LMask framework is designed to be general, making no prior assumptions about linearity or nonlinearity. Its strength lies in effectively handling hard constraints in any sequential decision-making process for machine learning-based routing problem solvers.
>
> > **Q1. Is TSL initialization the same as PIP?**
>
> Yes, the TSL initialization strategy in LMask is the same as that used in PIP, but we have optimized its implementation. Specifically, the PIP source code implementation requires extracting the current unvisited nodes, the updated unvisited nodes after one trial step, and their related data, all through indexing operations. In contrast, the LMask implementation first evaluates the constraints for all nodes, and only at the end uses a mask to exclude the influence of visited nodes on the final result. This approach is better suited for the backtracking mechanism, where different instances in a batch may have an inconsistent number of visited nodes at the same step. This implementation reduces the number of indexing operations and has proven to be more efficient in our experiments.
>
> > **Q2. What are Greedy-C and Greedy-L in Table 9?**
>
> We apologize for the ambiguity of the two baselines. Greedy-L and Greedy-C in Table 9 are greedy versions of Random-L and Random-C, respectively. Specifically, at each step, among all candidate nodes, Greedy-L selects the nearest available node, while Greedy-C selects the node that best satisfies the problem-specific constraints, i.e., the node with the earliest due time in TSPTW or the smallest draft limit in TSPDL.

---

### Meta-Review · Area_Chair_GkD9 · 2025-12-19

**Summary:**

This paper proposes LMask for constrained routing, using LazyMask with budgeted backtracking to improve feasibility, and introduces an embedding to incorporate refinement/backtracking information. Overall, the submission is solid and the empirical results on TSPTW/TSPDL are convincing, especially regarding feasibility and solution quality.
The rebuttal process addresses most concerns: it clarifies runtime/overhead, reports budget settings more clearly, adds ablations, and provides evidence that the approach is not tightly coupled to a single backbone. However, there are still concerns about 1）broader applicability beyond the selected problems, and 2) more systematic analysis of finite-budget failure cases and potential-set overestimation.
Given that most of the reviewers gave positive evaluations, I recommend accepting this paper as a poster, and suggest the author add more discussions of the broader applicability in the paper, if applicable.

**Reviewer Concerns:**

Most of the concerns have been addressed, while several remain, such as 1) generality beyond the two benchmarks (KHo7); 2) residual infeasibility and failure-mode analysis (KHo7); 3) novelty vs. PIP framing (nv24); 4) theory–practice gap (nv24/jBMF/V3Tu), though the theory relies on assumptions and does not fully translate into concrete, prescriptive design rules under finite budget.

**Reviewer Scores:**

Three out of four reviewers are inclined to maintain their positive evaluations, and one reviewer may potentially revise a negative rating to a positive one.

---

### Decision · Program_Chairs · 2026-01-26

Accept (Poster)